# FlexRank: Nested Low-Rank Knowledge Decomposition for Adaptive Model Deployment

**Riccardo Zaccone** [1] [2]   **Stefanos Laskaridis** [3] [†]   **Marco Ciccone** [4]   **Samuel Horváth** [2]

## Abstract

The growing scale of deep neural networks, encompassing large language models (LLMs) and vision transformers (ViTs), has made training from scratch prohibitively expensive and deployment increasingly costly. These models are often used as computational monoliths with fixed cost, hindering adaptive deployment across different cost budgets. We argue that nested components, ordered by importance, can be extracted from pretrained models and selectively activated within the available computational budget. To this end, our proposed FLEXRANK method leverages low-rank weight decomposition with nested, importance-based consolidation to extract submodels of increasing capabilities. Our approach enables a *"train-once, deploy-everywhere"* paradigm offering a graceful trade-off between cost and performance without training from scratch for each budget — advancing practical deployment of large models[1].

## 1. Introduction

Over recent years, the number of parameters and computational demands of modern networks have grown dramatically. Large Language (Vaswani et al., 2017) and Vision (Dosovitskiy et al., 2021) Transformer models now contain billions of parameters and require vast training corpora and compute budgets (Adler et al., 2024; Team et al., 2025; Grattafiori et al., 2024; Qiu et al., 2025). As these models continue to scale, training from scratch has become feasible only for a small number of well-resourced institutions. This has prompted the broader community to reuse publicly released pre-trained models and develop increasingly sophisticated methods for adapting them to downstream tasks (Hu et al., 2022; Liu et al., 2024; Wang et al., 2024; Li et al., 2024; Tastan et al., 2025).

Most adaptation strategies fall under the umbrella of parameter-efficient fine-tuning (PEFT). These methods modify a small auxiliary set of parameters while keeping the original model largely frozen. Although PEFT substantially reduces training costs, it leaves the computational structure of the backbone network unchanged. Consequently, model size and inference cost remain fixed, even when downstream applications could benefit from lighter or more variable configurations. This mismatch between adaptation flexibility and deployment rigidity becomes more pronounced when targeting environments with diverse hardware capabilities and different latency or memory budgets (Laskaridis et al., 2024; Wu et al., 2019; Zheng et al., 2025).

A natural direction for reducing inference cost is to compress the model itself. Two families of techniques have become particularly prominent. Quantization (Lin et al., 2024; Frantar et al., 2023; Xiao et al., 2023) reduces numerical precision to shrink memory footprint and improve throughput, but highest-quality results typically rely on quantization-aware training (QAT) (Liu et al., 2025; Chen et al., 2025; Ma et al., 2024), which requires modifying the training pipeline. Sparsity-based approaches (Sun et al., 2024; Frantar & Alistarh, 2023; Kurtić et al., 2023), instead, prune weights or induce structure via penalties, such as $\ell_1$ regularization. However, these methods also require retraining the full model to maintain performance and often depend on hardware, kernel support, or specific sparsity patterns to translate sparsity into real speedups (Mishra et al., 2021; PyTorch Documentation Team, 2025).

While highly effective, especially for memory-bound workloads, these approaches often fail to provide a flexible, smoothly varying set of model capacities, which are increasingly needed in heterogeneous deployment environments. For example, consumer device manufacturers typically offer a variety of platforms of varying capabilities that span across different tiers and generations, thus offering different compute and capacity dynamics. Additionally, model inputs vary in difficulty and can be handled by smaller model variants without compromising accuracy. However, training and maintaining submodels of different sizes is costly and cum-

---

[1]Code available at https://github.com/RickZack/FlexRank

[†]Work done independently of Amazon.  [1]Polytechnic of Turin [2]Mohamed bin Zayed University of Artificial Intelligence [3]Amazon Science, UK [4]Vector Institute. Correspondence to: Riccardo Zaccone <riccardo.zaccone@polito.it>.

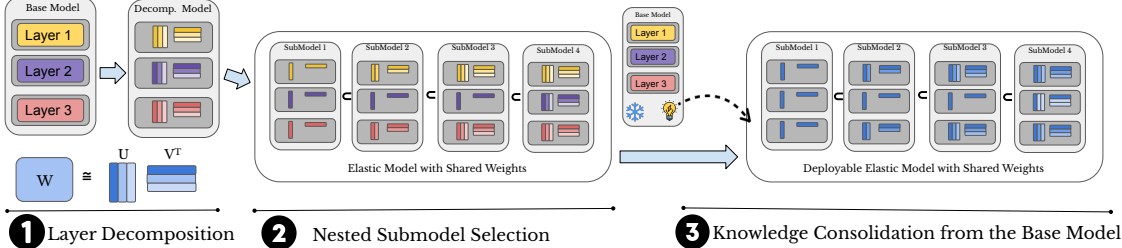

**1** Layer Decomposition     **2** Nested Submodel Selection     **3** Knowledge Consolidation from the Base Model

*Figure 1.* FLEXRANK takes as input a base model, which is first decomposed by factorizing each linear layer independently. Next, a global ordering is obtained via a dynamic programming subroutine that assumes additivity of errors across layers. This global ordering is then used to extract nested submodels of different sizes, which are stochastically refined through distillation from the base model.

bersome. This motivates exploring alternative mechanisms for inducing *model elasticity*.

Most current techniques construct elasticity through mechanisms such as joint training of several model sizes via architectural slicing of a pretrained network (Horváth et al., 2021; Cai et al., 2019; Devvrit et al., 2024) or dynamic routing (Cai et al., 2024; 2025). Yet the way these methods extract submodels frequently limits the quality of the resulting efficiency–performance trade-offs. First, many such methods predetermine a small set of submodel sizes, often chosen uniformly or heuristically, rather than discovering the configurations that truly lie on the Pareto frontier. Second, it is usually assumed that the pretrained model already contains viable subnetworks. Yet, these subnetworks may have never been encouraged to be performant during training and often compete for representation capacity.

To overcome these limitations, we propose FLEXRANK, a framework that decomposes the layers of any pretrained model into importance–ordered directional components and then efficiently searches and refines nested submodels that approximate the Pareto frontier. This two-stage process enables principled, nested elasticity without training multiple models from scratch or relying on architectural heuristics.

**Contributions.** We make the following contributions:

- We propose FLEXRANK, a rank-based elastic method that decomposes a pretrained model into nested, importance-ordered submodels within a single set of weights.
- We show that nested submodel training is key to Pareto-efficient low-rank elasticity, and combine this insight with a dynamic-programming procedure (DP) for selecting near-optimal rank configurations across budgets.
- We introduce a *"Gauge-Aligned Reparametrization"* (GAR), translating rank selection into practical inference savings, and extensively show that FLEXRANK improves accuracy–cost trade-offs across DNNs, ViTs, and LLMs.

## 2. Preliminaries

We first formally define the notion of an *elastic model* and discuss its desired properties. We then introduce the underlying objective of *elastic training*.

### 2.1. Elastic Models

Versatile deployment requires models that can adapt to varying hardware constraints without storing separate parameters for every possible configuration. We formalize this requirement as follows. Let $\mathcal{D}$ denote a data distribution and $\mathcal{B}$ a set of computational budgets corresponding to realizable configurations, *i.e.*, $\mathcal{B} = \{\beta_i\}_{i=1}^{N}$. A model is defined as a tuple $(f, \theta)$, where $f(\mathbf{d}; \theta)$ is a function parameterized by shared weights $\theta$ and $\mathbf{d} \sim \mathcal{D}$ denotes the input.

From a single set of shared parameters $\theta$, we derive a family of model realizations via $\mathcal{T}_\beta(\cdot)$, an algorithm-dependent transformation parameterized by the budget $\beta \in (0, 1]$. We denote by $\mathcal{R}(\cdot)$ and $\mathcal{C}(\cdot)$ the performance and cost operators, respectively, subject to the constraint $\mathcal{C}(f(\mathbf{d}; \mathcal{T}_\beta(\theta))) \leq \beta$. To illustrate, consider two common compression strategies. For sparsification, $\mathcal{B}$ represents sparsity ratios and $\mathcal{T}_\beta(\theta) = M_\beta \odot \theta$, where $M_\beta \in \{0, 1\}^d$ is a binary mask with density proportional to $\beta$. For quantization, $\mathcal{B}$ defines a set of relative bit-widths, and $\mathcal{T}_\beta$ is the corresponding quantization operator.

In this work, we focus on **low-rank approximations** of weight matrices. Specifically, each weight matrix $\{W_l \in \mathbb{R}^{m_l \times n_l}\} \subset \theta$ is factorized as $W_l = U_l V_l^\top$, with factors $U_l \in \mathbb{R}^{m_l \times r_l}$ and $V_l \in \mathbb{R}^{n_l \times r_l}$. The budget parameter $\beta$ controls the total number of preserved parameters via a transformation $\mathcal{T}_\beta$ that selects, for each layer, a subset of indices $\mathcal{S}_l \subseteq [r_l] = \{1, 2, \ldots, r_l\}$ and retains only the corresponding columns of $U_l$ and $V_l$. The subsets $\{\mathcal{S}_l\}$ are chosen across layers to satisfy a global budget constraint induced by $\beta$. We discuss how standard neural network layers can be parameterized within this framework in App. D.3.

More generally, $\mathcal{T}_\beta(\cdot)$ may be any transformation. Ideally, we would use the optimal transformation

$$\mathcal{T}_\beta^\star(\theta) \in \arg\min_{\mathcal{T}_\beta(\theta)} \mathbb{E}_{\mathbf{d} \sim \mathcal{D}} \left[ \mathcal{R}\big(f(\mathbf{d}; \mathcal{T}_\beta(\theta))\big) \right]. \quad (1)$$

However, solving (1) is largely intractable, as it entails a combinatorial optimization problem, different for each $\theta$. Consequently, additional assumptions are required to render the problem tractable. Ultimately, our goal is to find $\theta^\star$ that is Pareto optimal in the joint space of performance and cost.

**Definition 2.1** (Pareto Elastic Model). An elastic model $(f, \theta^\star)$ is *optimally elastic* if each configuration $f(\cdot; \mathcal{T}_\beta^\star(\theta))$ lies on the **Pareto front** $\mathcal{P}$ of the objective space $(\mathcal{R}, -\mathcal{C})$. Formally, the model $(f, \theta^\star)$ is optimally elastic iff for each $\beta \in \mathcal{B}$, there exists no $\tilde{\theta}$ and $\tilde{\mathcal{T}}_\beta$ such that:

$$\mathbb{E}_{\mathbf{d} \sim \mathcal{D}} \left[ \mathcal{R}\big(f(\mathbf{d}; \tilde{\mathcal{T}}_\beta(\tilde{\theta}))\big) \right] > \mathbb{E}_{\mathbf{d} \sim \mathcal{D}} \left[ \mathcal{R}\big(f(\mathbf{d}; \mathcal{T}_\beta^\star(\theta^\star))\big) \right].$$

Optimal elasticity characterizes an ideal weight-sharing regime in which a single parameter vector $\theta$ simultaneously encodes optimal representations for all budgets. This formulation decouples representation learning – encoded in the shared parameters $\theta$ – from budget-specific realization governed by the transformation $\mathcal{T}_\beta$. *Achieving this property constitutes the **central challenge** addressed in this work.*

### 2.2. Training Elastic Models

The Pareto elastic model is idealized and serves as a conceptual upper bound rather than an attainable objective; in practice, we therefore turn to a more tractable optimization formulation that approximates Pareto elasticity, defined as:

$$\hat{\theta}^\star \in \arg\min_\theta \sum_{\beta_k \in \mathcal{B}} \alpha_k \mathbb{E}_{\mathbf{d} \sim \mathcal{D}} \left[ \mathcal{L}(f(\mathbf{d}; \mathcal{T}_{\beta_k}^\star(\theta))) \right], \quad (2)$$

where $\mathcal{L}(\cdot)$ is a task-specific loss, and $\{\alpha_k > 0\}$ are coefficients prioritizing different budget regimes. Note that (2) defines an implicit bilevel optimization problem, since the optimal transformation $\mathcal{T}_\beta^\star$ itself depends on $\theta$. This problem is difficult to solve as gradient-based optimization cannot be directly applied due to the complexity of computing $\mathcal{T}_{\beta_k}^\star(\theta)$. Consequently, existing approaches typically either *(i)* optimize only the full model and then extract submodels from frozen parameters, or *(ii)* optimize multiple submodels for different budgets and select the best-performing ones post hoc. As we argue in Sec. 4, both strategies are suboptimal, motivating the need for a novel approach.

## 3. FlexRank

We are now ready to introduce FLEXRANK, a principled framework for low-rank knowledge decomposition that enables elastic model scaling. As previously discussed (Sec. 2.2), directly solving the resulting bilevel optimization is intractable for large models. We therefore focus on a practical and scalable setting in which a high-capacity pre-trained base model is available, an assumption well aligned with modern deployment pipelines (Adler et al., 2024; Team et al., 2025; Grattafiori et al., 2024; Qiu et al., 2025).

Specifically, FLEXRANK takes as input a pre-trained network $(f, \theta)$ and a calibration dataset $\mathcal{Z}$ [2]. Rather than jointly optimizing elastic submodels end-to-end, we leverage the

---

[2] In practice, this can be a representative subset of pretraining or downstream data (i.e. similar datamix), at the scale of $10^3$ samples.

base model to efficiently identify high-quality submodels that act as proxies for determining the transformation operator $\mathcal{T}_\beta^\star$. We make the key assumption that, although further optimization is required to obtain strong deployable models, the relative importance ordering induced by these submodels is preserved; that is, the optimal mask structure produced by $\mathcal{T}_\beta^\star$ is fixed, allowing subsequent optimization to focus solely on the parameters.

An overview of FLEXRANK is shown in Fig. 1 and Algorithm 1. The method comprises three stages. **1** **Layer Decomposition:** Each layer of the base model is independently decomposed via singular value decompositions, yielding an optimal per-layer factorization of ordered, nested components (Sec. 4.4). **2** **Nested Submodel Search:** Assuming ranking preservation of solutions under an additive error probe across layers (see App. C.3), we compute a global importance ordering by solving a dynamic program, yielding a collection of nested submodels across budgets. **3** **Knowledge Consolidation:** As the decomposed submodels ignore cross-layer dependencies, we refine all configurations through knowledge distillation (KD) from the base model, yielding deployable elastic models with shared weights.

---

**Algorithm 1** FLEXRANK

**Require:** Pretrained model $f(\cdot; \theta_{\text{orig}})$; calibration data $\mathcal{Z}$; training data $\mathcal{D}$; training budgets $\mathcal{B} = \{\beta_k\}_{k=1}^K$
**Ensure:** Elastic parameters $\theta$ and ordered Pareto front $\mathcal{M}^\star$

    LAYER DECOMPOSITION
1: **for all** layers $l = 1, \dots, L$ with weight $W_l$ **do**
2:     collect activations $X_l$ from $\mathcal{Z}$
3:     compute DataSVD factors $(U_l, V_l)$ by solving Eq. (3)
4: **end for**
5: initialize $\theta \leftarrow \{(U_l, V_l)\}_{l=1}^L$
    NESTED SUBMODEL SEARCH
6: **for all** layers $l = 1, \dots, L$ **do**
7:     $\mathcal{M}_l \leftarrow \{\mathbf{m}_r : r \in \mathcal{U}(r_l, K)\}$     ▷ truncate only layer $l$
8:     $\mathcal{Q}_l \leftarrow \{(\Delta c, \Delta e, r) : \mathbf{m}_r \in \mathcal{M}_l\}$
9:     **where** $\Delta c = \mathcal{C}(f, \theta) - \mathcal{C}(f, \mathcal{T}_{\mathbf{m}_r}(\theta))$
10:         $\Delta e = \mathcal{R}(f, \theta) - \mathcal{R}(f, \mathcal{T}_{\mathbf{m}_r}(\theta))$
11: **end for**
12: $\mathcal{M}^\star \leftarrow \text{DPRANKSELECTION}(\{\mathcal{Q}_l\}_{l=1}^L)$     ▷ Algorithm 2
    KNOWLEDGE CONSOLIDATION
13: $\widehat{\mathcal{M}} \leftarrow \text{SELECTPROFILES}(\mathcal{M}^\star, \mathcal{B})$     ▷ best budget profiles
14: **for** training steps $t = 1, \dots, T$ **do**
15:     sample $\mathbf{m}_t^\star \sim \widehat{\mathcal{M}}$ and minibatch $\mathbf{d} \sim \mathcal{D}$
16:     $\theta \leftarrow \text{OPT}\big(\theta, \nabla_\theta \mathcal{L}_{\text{KD}}\big(f(\mathbf{d}; \mathcal{T}_{\mathbf{m}_t^\star}(\theta)), f(\mathbf{d}; \theta_{\text{orig}})\big)\big)$
17: **end for**
18: **return** $\theta, \mathcal{M}^\star$
    DEPLOY EVERYWHERE
    **Input:** model $\theta$; Pareto front $\mathcal{M}^\star$; deployment budget $\beta$
19: $\{\mathbf{m}_\beta^\star\} \leftarrow \text{SELECTPROFILES}(\mathcal{M}^\star, \{\beta\})$
20: **for all** factorized layers $l = 1, \dots, L$ **do**
21:     $(\widehat{U}_l, \widetilde{V}_l) \leftarrow \text{GAR}(U_l, V_l, (\mathbf{m}_\beta^\star)_l)$     ▷ see Eq. (7)
22: **end for**
23: $\theta_\beta \leftarrow \{(\widehat{U}_l, \widetilde{V}_l)\}_{l=1}^L$
24: **return** $\theta_\beta$

---

## 3.1. Layer Decomposition

To initialize the shared decomposed parameters $\theta$, we first perform a layer-wise factorization of the pretrained weights $\{W_l\}$. We seek an initialization that preserves the functional behavior of the original model and admits a closed-form solution, which we refer to as *DataSVD*. Specifically, we compute low-rank factors $U_l$ and $V_l$ by minimizing the output reconstruction error

$$\min_{U_l, V_l} \ \mathbb{E}_{\mathbf{x}_l \sim \mathcal{X}_l} \left[ \left\| (W_l - U_l V_l^\top) \mathbf{x}_l \right\|_2^2 \right], \qquad (3)$$

where $\mathbf{x}_l$ denotes the input activations to layer $l$ and $\mathcal{X}_l$ is the corresponding distribution. In practice, (3) can be approximated by sampling a matrix $\mathbf{X}_l \in \mathbb{R}^{n_l \times N}$ containing activation vectors collected from a calibration dataset, with $N$ chosen large enough to capture the principal directions of the data. As we show in App. C.1 and the space complexity can be made independent of $N$, scaling as $\mathcal{O}(n_l^2)$.

Solving the objective via SVD crucially induces a natural ordering of rank components within each layer, which enables a tractable global selection of components across layers via dynamic programming, as described next.

*Remark* 3.1. Data-aware decompositions of this form are not new, and they have been in prior work (*e.g.*, (Chen et al., 2021)). However, in our method, it serves only as initialization to extract candidate submodels from a pretrained network. Importantly, this initialization alone is not sufficient to recover good submodels, as we show in Sec. 5.

## 3.2. Approximating the Pareto Front

Following the notation of Sec. 2.1, for a given budget $\beta_k \in \mathcal{B}$, the transformation operator $\mathcal{T}_{\beta_k}$ selects a rank $r_{k,l} \leq r_l$ for each layer $l$, such that the overall budget constraint is satisfied. We denote the corresponding configuration vector by $\mathbf{m}_k = \{r_{k,l}\}_{l=1}^L$. We further denote by $\mathcal{T}_{\mathbf{m}_k}$ the transformation induced by $\mathbf{m}_k$. To obtain an optimally elastic model, we seek a set of configurations $\mathcal{M} := \{\mathbf{m}_k\}_{k=1}^K$ that solve the following problem

$$\min_{\{\mathbf{m}_k\}_{k=1}^K} \sum_{k=1}^K \mathbb{E}_{\mathbf{d} \sim \mathcal{D}} \left[ \mathcal{L}\big( f(\mathbf{d}; \mathcal{T}_{\mathbf{m}_k}(\theta^0)) \big) \right], \qquad (4)$$

where $\theta^0$ denotes the shared parameters obtained after layer-wise decomposition. Importantly, we optimize only over the masks $\{\mathbf{m}_k\}$. While $\theta^0$ is not a final deployable model, we assume it is ***sufficient to identify configurations whose relative optimality is preserved throughout subsequent training***.

**Nestedness.** We further impose a nestedness constraint $\mathbf{m}_{k-1} \preceq \mathbf{m}_k$ in Eq. (4). This is critical for limiting weight-sharing interference, as inconsistent rank selections across layers would prevent the shared parameters $\theta$ from converg-

ing to a coherent representational hierarchy. We provide theoretical evidence for this in a simplified setting in Sec. 4.

Identifying the optimal set $\mathcal{M}$ is combinatorial: even with $K$ candidate budgets across $L$ layers, the search space contains $K^L$ submodels. Since the initial decomposition is performed independently per layer, we implicitly assume that layers are approximately independent under $\theta^0$. Accordingly, we assume that the ranking of possible submodels based on the combined low-rank approximations across layers is preserved under an additive probe, *i.e.* we can predict the ranking of solutions in the search space by assuming the errors are additive. This is a strong but standard assumption (Hubara et al., 2021), which we validate in an exhaustively searchable setting in App. C.3. Crucially, it enables an efficient dynamic programming solution with complexity $\mathcal{O}(L \cdot K)$ (see Algorithm 2). The resulting procedure is summarized in App. C.2.

## 3.3. Elastic Training Objective

Once the set of optimal configurations $\mathcal{M}^\star = \{\mathbf{m}_k^\star\}_{k=1}^K$ is identified, we fix these mappings for each budget level $\beta_k \in \mathcal{B}$. Thus, for the remainder of the training phase, the operator $\mathcal{T}_{\mathbf{m}_k^\star}$ is fixed independent of the current $\theta$ according to the obtained rank assignments in $\mathbf{m}_k^\star$. We optimize $\theta$ by minimizing the distillation loss ($\mathcal{L}_{\mathrm{KD}}$) between the elastic submodels and the original, non-decomposed pretrained model $f(\cdot; \theta_{\mathrm{orig}})$. Since a strong pretrained base model is available, training from teacher logits provides a substantially richer supervision signal than ground-truth labels. We define the loss for the $k$-th budget level as:

$$\ell_k(\theta) = \mathbb{E}_{\mathbf{d} \sim \mathcal{D}} \left[ \mathcal{L}_{\mathrm{KD}}\big( f(\mathbf{d}; \mathcal{T}_{\mathbf{m}_k^\star}(\theta)), f(\mathbf{d}; \theta_{\mathrm{orig}}) \big) \right]. \quad (5)$$

Therefore, the final objective becomes:

$$\min_{\theta \in \mathbb{R}^d} \sum_{k=1}^K \alpha_k \ell_k(\theta), \quad \text{s.t.} \ \sum_{k=1}^K \alpha_k = 1, \quad (6)$$

which can be efficiently solved by standard gradient-based optimization, where we sample $\ell_k$ proportionally to $\alpha_k$.

## 3.4. Recovering the True Pareto Front

We empirically validate the central approximation underlying FLEXRANK that nested configurations identified from a fixed layer-wise optimal initialization are sufficient to recover Pareto-optimal submodels after training. In a setting tractable to allow complete enumeration of the solution, we actually verify that FLEXRANK recovers the *true* Pareto front obtained by independently training all possible submodels. We therefore consider a four-layer network (two CNNs and two MLPs trained on MNIST, with $K = 10$ rank levels per layer, yielding $K^L = 10{,}000$ possible submodels. This allows us to exhaustively evaluate the attainable Pareto front by independently training all submodels more details on this experimental setting are reported in App. D.1.

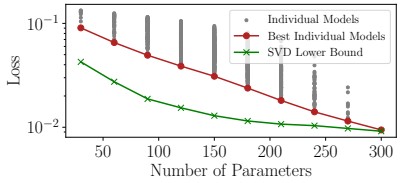

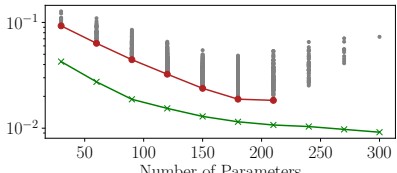

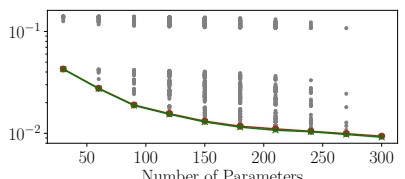

(a) Post-Training Selection (PTS)   (b) All Submodel Learning (ASL)   (c) Nested Submodel Learning (NSL)

*Figure 2.* **Nested trained submodels are Pareto Elastic:** Comparison of the considered submodels training strategies on the synthetic setting described in App. D.1. Blue points visualize all 1023 submodels, the red line represents the best models and the green one the true Pareto Front. The difference between the red and green lines is the best submodel optimality gap as per Eq. (9), and is zero only for NSL.

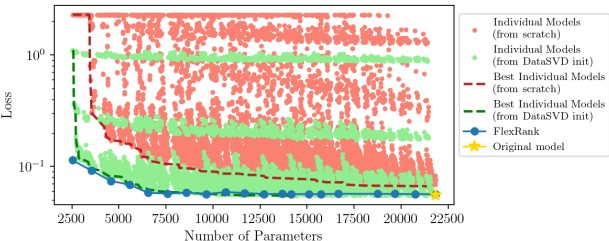

*Figure 3.* **FLEXRANK recovers the true Pareto Front in DNNs:** points represent independently trained nested submodels, starting from (i) a random weights (red) or (ii) from the DataSVD (green) of a pretrained model (yellow star), with best models highlighted with dashed lines. At convergence, FLEXRANK recovers the (in advance unknown) Pareto front within a single set of shared weights.

Nestedness and weight sharing make optimization more constrained, but impose a hierarchy of shared rank components that transfers knowledge across budgets: larger submodels refine representations reused by smaller ones, while smaller submodels keep shared components compact and reusable. Fig. 3 supports this interpretation: FLEXRANK recovers the Pareto front of independently trained DataSVD-initialized models while outperforming models trained from scratch.

### 3.5. Gauge-Aligned Reparametrization (GAR)

We introduce an efficient reparametrization of the factorized weights which, once a target rank $r$ is fixed at inference, reduces the cost of matrix–vector multiplication to $\mathcal{O}((m + n - r)r)$, which is strictly less than the $\mathcal{O}(mn)$ FLOPs required for dense multiplication for any $r < \min(m, n)$. The technique comprises exploiting the non-uniqueness of the $(U, V)$ factorization, in order to avoid storing and multiplying a dense $(r \times r)$ block. In practice, defining the "gauge" $G = U_{1:r,:}^{-1}$, it follows that:

$$UV^\top = \underbrace{(UG)}_{\tilde{U} \in \mathbb{R}^{m \times r}} \underbrace{(G^{-1}V^\top)}_{\tilde{V}^\top \in \mathbb{R}^{r \times n}}, \quad \tilde{U} = \left[ \underbrace{I_r}_{r \times r} \quad \underbrace{\hat{U}}_{(m-r) \times r} \right]^\top, \quad (7)$$

where the identity $I_r$ does not need to be stored or multiplied explicitly. Using the GAR form $(\hat{U}, \tilde{V})$ reduces the standard inference cost of a rank-$r$ factorization, scaling linearly with $r$ and enabling computational gains without aggressive rank reduction. The matrix $G$ is computed once per layer after

rank selection via a matrix inversion, costing $\mathcal{O}(r^3)$ FLOPs, which is negligible compared to SVD calculation.

As GAR is general to full rank matrices, we do not consider it exclusive to FLEXRANK, and apply it uniformly across all rank-based methods in our experiments (see Remark 5.1).

## 4. The Need for Nestedness

In this section, we discuss how to formulate elastic training in the rank space to obtain Pareto-optimal submodels. For linear models we prove: *(i)* why training only the full model does not recover optimal submodels; and that *(ii)* training all possible submodels leads to degradation of the Pareto Front. We finally prove the core result our method is based on: Pareto-optimal submodels are found by training only "nested" submodels. Proofs are deferred to App. B.

### 4.1. Setup

Consider a linear model represented by a matrix $M \in \mathbb{R}^{m \times n}$, parameterized as $M = UV^\top$ with $U \in \mathbb{R}^{m \times k}$, $V \in \mathbb{R}^{n \times k}$, and $k = \min(m, n)$. Let $M^\star$ be the optimal solution, and assume it has SVD $P\Sigma Q^\top$ satisfying: $\Sigma = \mathrm{diag}(\sigma_1, \ldots, \sigma_k)$, $\sigma_i > \sigma_{i+1} > 0$ $\forall i < k$. Let $\Pi_{S_r} = \mathrm{diag}(s_1^r, \ldots, s_k^r)$, where $s_i^r = \mathbf{1}(i \in S_r)$. Then, Eq. (2) under low-rank transformation is equivalent to:

$$\min_{U, V, \{S_r\}_{r=1}^k} \frac{1}{k} \sum_{r=1}^k \|U\Pi_{S_r}V^\top - M^\star\|_F^2. \quad (8)$$

For $r \in [k]$, the best rank $r$ appproximation of $M^\star$ is equal to the (unique) truncated SVD $A_r = \sum_{i=1}^r \sigma_i \, p_i q_i^\top$, where $p_i, q_i$ denote the $i$-th columns of $P, Q$. By the Eckart–Young–Mirsky theorem, it follows that the Pareto front is the set $\{A_r\}_{r=1}^k$. The best submodel optimality gap is then:

$$\mathcal{E}(U, V, r) := \min_{S_r \subseteq [k]} \left\|U\Pi_{S_r}V^\top - A_r\right\|_F^2. \quad (9)$$

This represents the lower bound on the reconstruction error of submodels learned by any algorithm. In particular, in the rest of the section, we assume we have access to the best selection indices $S_r$, which is always theoretically possible by exhaustively exploring the whole search space.

## 4.2. Why Post-Training Selection (PTS) Fails

One simple approach to solve Eq. (8) is based on a two-stage procedure: firstly, the full model is parametrized in the form $(U, V)$ and trained from scratch, by only optimizing

$$\min_{U,V} \|UV^\top - M^\star\|_F^2. \tag{10}$$

Secondly, the best selection indices $S_r$ are found for all $r \in [k]$. We call this algorithm Post-Training Selection (PTS), since it is based on a post-hoc selection of singular vectors after regular training. PTS provably does not lead to *optimally* elastic models, as it is always possible to find a model that performs better at any reduced parameter budget.

**Theorem 4.1.** *Let* $\mathcal{M} := \{(U, V) : UV^\top = M^\star\}$ *be the set of global minimizers of* (10). *Then, for each* $r < k$, *the set* $\mathcal{M}_r := \{(U, V) \in \mathcal{M} : \mathcal{E}(U, V, r) = 0\}$ *has Lebesgue measure zero relative to* $\mathcal{M}$.

The theorem proves that the chance that any algorithm that only operates on (10) would find global minimizer of (8) is zero. Simulations in Fig. 2(a) confirm this phenomenon.

## 4.3. All-Subspaces Learning Degrades the Pareto Front

The main insight from Thm. 4.1 is that training only the full model is not enough to obtain an optimal solution across all ranks. Consequently, it is natural to consider training all possible submodels and then choose the best ones. We call this All-Subspaces Learning (ASL), where the objective is

$$\min_{U,V} \frac{1}{2^k - 1} \sum_{S \subseteq [k] : S \neq \emptyset} \|U\Pi_S V^\top - M^\star\|_F^2. \tag{11}$$

Similar to PTS, this approach also fails.

**Theorem 4.2** (ASL has strictly positive submodel gap)**.** *Let* $(U, V)$ *be any minimizer of* (11), *and let* $\lambda = \frac{1}{k}\|UV^\top\|_\star$. *Then for every* $r \in \{1, \ldots, k\}$, *the following holds*

$$\mathcal{E}(U, V, r) \geq \frac{1}{k}\left(r\lambda - \sum_{i=1}^{r}\sigma_i\right)^2.$$

The above theorem implies that for a generic matrix $M^\star$ with non-identical singular values, the optimality gap is strictly positive for some $r$. This is also confirmed with numerical simulations presented in Fig. 2(b). The failure of ASL stems from a multi-objective conflict: different submodels corresponding to the same rank $r$ compete for representational capacity. Each submodel attempts to converge to the optimal $r$-rank approximation $A_r$, but in doing so, it disrupts the global parameters and shifts the overall objective. We provide a simple example of this "interference" of submodels objectives in Corollary B.8.

## 4.4. Nested Learning

To address the prior issues, we propose Nested Subspace Learning (NSL). The key intuition is to design the training objective so that the resulting parameters naturally inherit the prefix structure of the Pareto front, leading to:

$$\min_{U,V} \frac{1}{k} \sum_{r=1}^{k} \|U\Pi_{[r]}V^\top - M^\star\|_F^2. \tag{12}$$

Unlike ASL, NSL optimizes exactly one submodel for each rank $r \in [k]$ and enforces sub-objective compatibility by construction. Since each sub-objective $r$ targets the $r$-rank approximation $A_r$, the nested structure ensures that the additional $(r+1)$-th column only needs to learn the residual $A_{r+1} - A_r$. Consequently, NSL recovers the Pareto front.

**Theorem 4.3** (NSL preserves nested minimizers)**.** *Let* $(U, V)$ *a minimizer of* (12), *then* $\forall r \in [k] : \mathcal{E}(U, V, r) = 0$.

Simulations in Fig. 2(c) align with the theory: nested training successfully recovers the Pareto Front $\{A_r\}_{r=1}^{k}$.

# 5. Experiments

**Training.** We consider both NLP and CV tasks: for the former, we employ GPT-2 and three recent models of the Llama family (3.2-1B, 3.2-3B, and 3.1-8B) (Grattafiori et al., 2024). The calibration dataset used for FlexRank is FineWebEdu-10BT (Penedo et al., 2024). For CV, we employ the recent DINOv3 ViT models (Siméoni et al., 2025), ranging from the ViT-L/16 up to the ViT-7B/16, and choose image classification on ImageNet1K as the downstream task.

**Evaluation.** For GPT-2, we show results on the evaluation loss on a held-out split of the proxy dataset, as is common in the literature (Genzel et al., 2025). For Llama models, we evaluate the zero-shot accuracy of models on commonsense datasets commonly used in the literature using the lm-eval-harness tool (Gao et al., 2024), while for CV tasks, we evaluate on the validation split of ImageNet1K. Additional details on training and evaluation are provided in App. D.2.

**Comparisons.** Most low-rank compression methods select ranks from a per-layer SVD, either computed directly on the weights or informed by activations (*e.g.* ASVD (Yuan et al., 2023) and $A^3$ (Wong et al., 2025)). We therefore benchmark SVD and DataSVD with our DP search, isolating the effect of nested submodel training. A second class of methods performs additional optimization to recover the error introduced by compression, either by updating the full weights, as in DRONE (Chen et al., 2021), or by training LoRA adapters, as in SVD-LLM (Wang et al., 2025b). Within this family, we compare against ACIP (Genzel et al., 2025), the current state-of-the-art low-rank elastic method. ACIP prunes an SVD-decomposed pretrained model with frozen base weights, jointly optimizing LoRA adapters and pruning scores that define the submodels.

*Remark* 5.1. For all rank-based approaches, including the baselines, we apply GAR (Sec. 3.5) after rank selection. Crucially, this is why the reported inference-time relative parameter counts remain equal or lower than full model's.

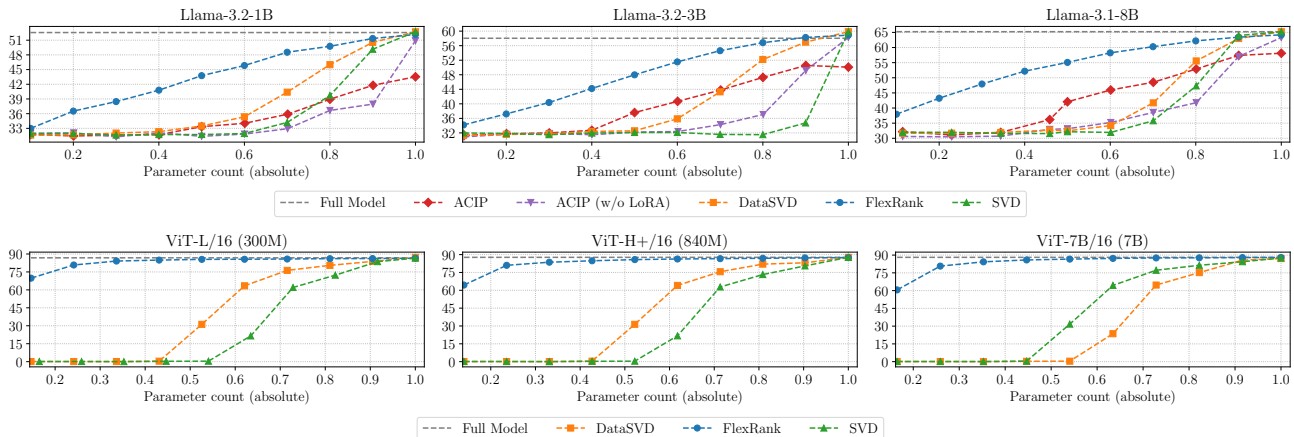

*Figure 4.* **FLEXRANK has the most graceful performance degradation across parameter budget:** (top) Average downstream task accuracy over commonsense downstream datasets from `lm-eval-harness` and (bottom) classification accuracy on the evaluation split of ImageNet1K. In this latter case, the performance gap remains within a 5% margin w.r.t. the full model even pruning up to 70%.

**Comparison at matched training budget.** We consider each competitor algorithm at their best tuning/budget. Due to the very different nature of approaches in the literature, with some methods just training small adapters (Genzel et al., 2025; Ma et al., 2023), this is the arguably the fairest choice. This holds both for lightweight ones (like LLM-PRUNER and ACIP) and for methods that are much heavier than FLEXRANK, such as LAYERSKIP, which has similar training complexity to ours but has been trained on $839B$ tokens (Elhoushi et al., 2024) (*i.e.*, $167\times$ our $5B$ token budget). In addition, in Fig. 5 we include a non-elastic baseline where we independently train the same submodels selected by FLEXRANK, starting from the same DataSVD initialization and using the same overall budget as FLEXRANK (*i.e.*, 10 models, each trained with 10% of the total budget).

### 5.1. Main Results

As shown in Fig. 4, methods based solely on SVD decomposition already degrade sharply after removing 20% of the parameters. This is consistent with the limitations of post-training selection discussed in Sec. 4.2: even when each layer admits an optimal SVD factorization, the resulting decomposition is not necessarily *globally* optimal, so even ideal submodel selection can incur a substantial drop.

For ACIP, Fig. 4-(top) shows that adding shared trainable parameters to compensate for compression error yields limited gains and can hinder full-budget recovery. This is connected to ASL-like dynamics (Sec. 4.3), where adapters compete for the representational capacity lost during compression. Without adapter training, ACIP reduces to a PTS-style method and recovers full-budget performance. This complements findings in the literature that adapters are often sufficient to recover from compression error in a single model and challenges the belief that such a simple approach could be as effective for elastic models.

For Vision Transformer results are even stronger: Fig. 4-(bottom) shows compressing to 30% of the original model size remains close to full-model performance. We attribute this partly to ImageNet1K being substantially smaller than FineWebEdu, allowing more epochs within the same compute budget. Fig. 5 includes a non-elastic strong baseline, where FLEXRANK submodels are independently trained from the same initialization and at matched budget, leading to 10 separate sets of weights. FLEXRANK slightly outperforms those models on average; we hypothesize that this is because it leverages weight sharing, enabling smaller submodels to help optimize larger ones.

**Comparison with other compression families.** The focus of this work is to advance rank-based elastic compression, because it provides a natural importance ordering through "SVD-like decompositions", which in turn yields a principled route to nested submodel construction. For this reason, prior methods in that regime are the most direct and informative baselines. However, low-rank compression is not the only possible route to elasticity: compression axes such as pruning, depth elasticity, or quantization are complementary rather than mutually exclusive.

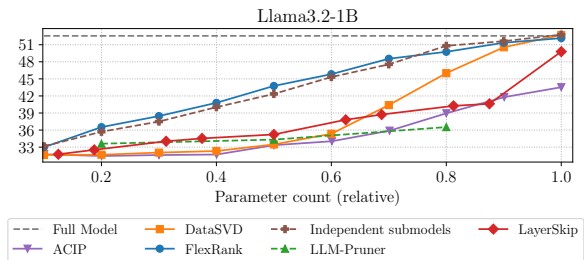

*Figure 5.* **FLEXRANK is competitive beyond rank-based approaches.** Comparison with LLM-PRUNER, LAYERSKIP and a baseline that trains the same FLEXRANK submodels independently at equal total budget. Dashed lines denote non-elastic methods.

To provide broader context, in the Fig. 5 we additionally compare against representative methods from other families, including LLM-PRUNER (Ma et al., 2023) (structured pruning) and LAYERSKIP (Elhoushi et al., 2024) (variable depth). On Llama-3.2-1B, these results suggest that FLEXRANK is competitive beyond rank-based approaches.

## 5.2. Post-Adaptation Performance

FLEXRANK submodels can be further finetuned and employed in downstream applications. We show this for math and code domains by following the common practice of adding lightweight LoRA adapters on individual submodels (additional details in App. D.2). Tab. 1 shows that, in both domains, the submodels achieve meaningful performance and exhibit graceful degradation across constrained budgets.

*Table 1.* **FLEXRANK submodels can be finetuned with LoRA:** average accuracy over math and code domains, at varying elastic sizes. "Base" denotes the performance of the original model.

| Relative Size | Llama-3.2-1B | | Llama-3.2-3B | |
|---|---|---|---|---|
| | Math | Code | Math | Code |
| Base | $25.69_{\pm0.99}$ | $18.33_{\pm2.35}$ | $40.56_{\pm1.16}$ | $36.78_{\pm2.95}$ |
| $1\times$ | $25.03_{\pm0.98}$ | $18.63_{\pm2.35}$ | $40.49_{\pm1.12}$ | $33.76_{\pm2.90}$ |
| $0.8\times$ | $20.48_{\pm0.95}$ | $9.30_{\pm1.83}$ | $32.95_{\pm1.08}$ | $22.59_{\pm2.58}$ |
| $0.6\times$ | $15.70_{\pm0.73}$ | $3.34_{\pm1.14}$ | $23.84_{\pm0.96}$ | $6.67_{\pm1.58}$ |
| $0.4\times$ | $13.58_{\pm0.63}$ | $1.22_{\pm0.61}$ | $15.54_{\pm0.71}$ | $2.03_{\pm0.88}$ |

## 5.3. Ablations

**Analysis of model profiles** Fig. 6 shows the compression factors corresponding to four submodels, from bigger to smaller, on GPT-2. As can be seen, not all components of the network are uniformly truncated to meet each parameter budget $\beta_k$, indicating that our DP-based algorithm accounts for the importance of each module and reduces its rank accordingly. Interestingly, the third heatmap shows that the c_proj of the central attention layers seems particularly important, as the algorithm truncates them only at the last.

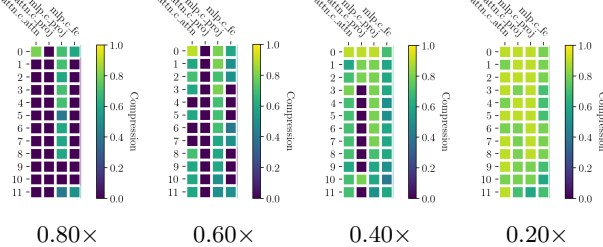

$$0.80\times \qquad 0.60\times \qquad 0.40\times \qquad 0.20\times$$

*Figure 6.* **FLEXRANK takes into account parameter importance (GPT-2):** Heatmaps of compression ratio of model components over submodels labeled by relative size w.r.t. the full model.

**The limits of SVD-based initialization.** In Fig. 7(a), we study the effect of activation sample size on the DataSVD initialization of FLEXRANK. A few hundred samples suffice

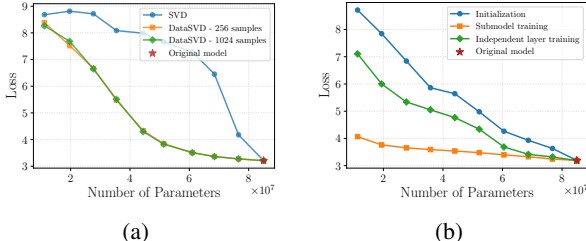

(a) (b)

*Figure 7.* **Limits of SVD initialization, need for submodel training (GPT-2).** (a) Superimposed green and orange curves show DataSVD converging with a few hundred samples. (b) Independent layer training (green) is ineffective, while end-to-end submodel training (orange) consolidates local into global nestedness.

to capture the dominant activation directions, as using more than 128 samples yields no visible gains. This suggests that, in DNNs, other sources of error dominate, limiting the impact of increasingly accurate per-layer decompositions.

**The need for training submodels** In Fig. 7(b), we analyze the residual errors after decomposing pretrained layers via DataSVD. If a "global SVD" decomposition across the entire architecture were possible, distillation would be unnecessary, and Pareto-optimal models would coincide with SVD truncations. However, in deep non-linear networks, performance degradation arises from (i) non-linear activation functions and (ii) complex information flow across depth. To isolate the former, we independently train each layer during distillation, enabling adaptation to its nonlinearity. The persistently poor performance shows that inter-layer information flow induces non-trivial dynamics, requiring end-to-end training to consolidate *local* (per-layer) nestedness into *global* nestedness.

**The importance of submodel sampling.** To assess the need for nested submodel training, we compare FLEXRANK with a baseline that trains a single submodel for the full budget. As shown in Fig. 8, independently trained submodels perform well only at their target budget and degrade markedly elsewhere, mirroring the PTS behavior in Sec. 4.2 and confirming that SGD alone does not support graceful rank reduction. In contrast, FLEXRANK matches the best independently trained submodel at each budget, indicating effective parameter sharing without interference.

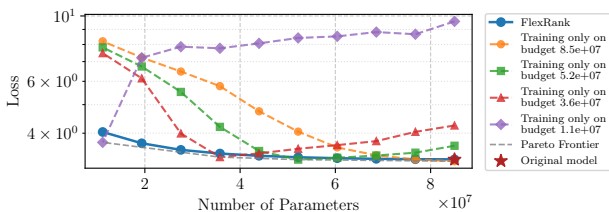

*Figure 8.* **Joint submodel training is essential for elasticity (GPT-2):** independently trained submodels lack nested structure and degrade across parameter budgets.

## 6. Related Work

**Model compression.** Deep neural networks typically show varying degrees of redundancy in their parametric knowledge. This overcommitment of representational capacity can manifest both in terms of numerical representation as well as knowledge superposition. Therefore, various quantization (Lin et al., 2024; Frantar et al., 2023; Xiao et al., 2023; Liu et al., 2025; Chen et al., 2025; Ma et al., 2024) and pruning techniques (Sun et al., 2024; Frantar & Alistarh, 2023; Kurtić et al., 2023) have been successfully applied to compress such models. However, these often requires access to the training set and/or specific hardware and kernel implementations to take advantage of the reduced compute.

**Low-rank methods.** Another approach is to leverage factorization methods to compress large models, without the need to alter dimensionality or hardware-awareness. Traditional approaches (Hsu et al., 2022; Wang et al., 2021; Horváth et al., 2024) have typically inherited a training-aware factorization workflow, which requires exposure to the full training process and datasets, which may not be tractable to many in the era of LLMs. As such, various techniques have been proposed for compressing pretrained LLMs (Chen et al., 2021; Wang et al., 2025b; Yuan et al., 2023; Genzel et al., 2025; Wong et al., 2025) by means of decomposition and truncated reparametrization. Common among various of these methods is the activation-informed decomposition, with many requiring additional training steps to recover lost accuracy (Chen et al., 2021; Wang et al., 2025b; Yuan et al., 2023). However, very few (Genzel et al., 2025) offer the ability of multi-model extraction, and less more so under a nested structure (see Tab. 2 for extended comparison).

**Flexible networks.** Another family of techniques aims at producing many models from a common backbone (or supernet (Cai et al., 2019)), which are usually optimized together. The selected/sampled submodels can operate at different depths (Fan et al., 2020; Raposo et al., 2024), widths (Yu et al., 2018; Devvrit et al., 2024; Horváth et al., 2021; Yu & Huang, 2019) or sub-architectures (Cai et al., 2019; 2024; 2025; Horváth et al., 2024) and can target inputs of varying difficulty (Cai et al., 2025), different target devices (Horváth et al., 2021; Lee et al., 2024) or even be used as draft models in speculative decoding (Elhoushi et al., 2024). Concurrently, Rauba & van der Schaar (2026) study rank-based elasticity: key differences lie in our superior decomposition (important for knowledge consolidation), submodel selection approximating the Pareto front, and in our GAR reparameterization (Sec. 3.5), crucially enabling much improved submodels without aggressive rank reductions.

To the best of our knowledge, FLEXRANK is the first work to lay the theoretical foundation for nested submodel training and to obtain flexible models at scale by operating in the factorized space.

## 7. Discussion & Limitations

**Optimization and adaptivity.** While FLEXRANK consistently improves the accuracy–cost trade-off over existing baselines, its performance is still constrained by the quality of the training set and the amount of consolidation training. Longer training, more diverse data, or optimization methods better tailored to jointly adapting nested submodels may further improve the Pareto frontier (Jordan et al., 2024). In addition, although FLEXRANK naturally enables budget-conditioned or input-adaptive inference, we do not evaluate adaptive routing policies in this work and leave this direction for future study.

**Training efficiency.** Algorithmically, each FLEXRANK training step is comparable to standard distillation, but on fewer tokens and with an elastic factorized model. Since the factors span the full rank of the original matrices during training, they introduce roughly a $2\times$ memory overhead and preclude fused dense kernels, making high-rank training about $2\times$ slower than a single dense forward. However, for elastic deployment, training cost is often secondary to obtaining many deployable submodels without retraining from scratch (Cai et al., 2019), as long as training the elastic model is comparable in terms of training effort (Yu et al., 2018). For example, Llama-3.2-1B was trained from the pretrained 8B model on a curated 9T-token corpus, whereas FLEXRANK uses only 5B tokens, making this cost appealing when amortized over many inference budgets.

**Inference efficiency.** At inference time, theoretical savings translate more directly into speedups. Our measurements in App. D.4 show that, for sufficiently large matrices and sequences where computation is the bottleneck, the GAR form (Sec. 3.5) closes much of the gap between naive low-rank factorization and dense kernels, with forward cost closely following the theoretical predictions.

## 8. Conclusion

In this work, we introduced FLEXRANK, a rank-based elastic method that decomposes a pretrained model into nested, importance-ordered components within a single set of weights. Combined with an efficient search and refinement procedure, FLEXRANK identifies submodels near the performance-latency Pareto frontier, without training multiple independent models or relying on architectural heuristics. Across architectures, FLEXRANK yields smoother accuracy degradation and improves the efficiency–performance trade-off over existing approaches, enabling adaptive deployment over diverse hardware and workloads.

## Impact Statement

This paper presents work aimed at advancing the field of Machine Learning. There are many potential societal consequences of our work, none of which we feel must be specifically highlighted here.

## Funding

Riccardo Zaccone declares that financial support was received for the research, authorship, and/or publication of this article. This study was carried out within the project FAIR - Future Artificial Intelligence Research - and received funding from the European Union Next-GenerationEU [PIANO NAZIONALE DI RIPRESA E RESILIENZA (PNRR) – MISSIONE 4 COMPONENTE 2, INVESTIMENTO 1.3 – D.D. 1555 11/10/2022, PE00000013 - CUP: E13C22001800001]. This manuscript reflects only the authors' views and opinions, neither the European Union nor the European Commission can be considered responsible for them. A part of the computational resources for this work was provided by hpc@polito, which is a Project of Academic Computing within the Department of Control and Computer Engineering at the Politecnico di Torino (http://www.hpc.polito.it). We acknowledge the CINECA award under the ISCRA initiative for the availability of high-performance computing resources. This work was supported by CINI.

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

# Appendix Contents

# A. Additional Discussion

## A.1. Relation to Other Flexible Non-Factorized Methods

We briefly discuss two recent works, FLEXTRON (Cai et al., 2024) and LLAMAFLEX (Cai et al., 2025), which are relevant in spirit but orthogonal to the focus of this paper. We do not compare against these methods experimentally, as they do not address low-rank approximation of weight matrices, which is the central mechanism studied in FLEXRANK. Moreover, to the best of our knowledge, neither work has released an implementation, and LLAMAFLEX in particular trains on proprietary data, limiting direct comparison.

Both FLEXTRON and LLAMAFLEX construct elastic Transformer models by varying architectural components, including the number of Transformer blocks, the hidden dimension size, the intermediate MLP dimension, and the number of attention heads. In contrast, FLEXRANK operates entirely in the factorized parameter space, producing elastic submodels by truncating low-rank representations while preserving the original architecture. Extending FLEXRANK to reason over architectural elasticity would be an interesting direction for future work.

From an optimization perspective, these methods are also closely related to the training paradigms analyzed in Sec. 4. In particular, LLAMAFLEX can be viewed as an instance of all-submodel learning (ASL), where multiple elastic configurations are optimized simultaneously without enforcing global nestedness across configurations. Similarly, FLEXTRON resembles a post-training selection (PTS) approach, in which a large super-network is trained, and submodels are selected afterward via routing decisions. Our theoretical results suggest that both strategies can lead to suboptimal Pareto fronts in the absence of fully nested training objectives.

These connections suggest promising future directions. On the one hand, it would be interesting to investigate whether enforcing global nestedness, as proposed in FLEXRANK, could further improve methods such as LLAMAFLEX, which currently enforce nestedness only at the component level. On the other hand, both FLEXTRON and LLAMAFLEX rely on learned routers for subnetwork selection, whereas FLEXRANK employs a deterministic dynamic programming procedure. Understanding how router-based selection compares to, or could be combined with, DP-based Pareto selection is an open and compelling direction for future work.

*Table 2.* Comparison of prior Transformer compression methods from the perspective of nested low-rank decomposition.

| Method | Decomposition | Rank Selection | Target Arch. | Acc. Compensation | Gradient-Free | Nestedness | Train-once, deploy-everywhere |
|---|---|---|---|---|---|---|---|
| Naive SVD | Weight SVD | Manual | Any linear | ✗ | ✓ | ✗ | ✗ |
| FWSVD (Hsu et al., 2022) | Fisher-weighted SVD | $r = \lfloor 0.33\min(N, M) \rfloor$ | Any linear | ✗ | ✗ | ✗ | ✗ |
| DRONE (Chen et al., 2021) | Data-informed SVD | Greedy layer-by-layer | Any linear | 1 epoch retrain | ✗ | ✗ | ✗ |
| ASVD (Yuan et al., 2023) | Activation-scaled SVD | Layer-wise calibration | Any linear | ✗ | ✓ | ✗ | ✗ |
| SVD-LLM (Wang et al., 2025b) | Whitened activations informed SVD | $r = \frac{NM}{N+M}(1 - R_w)$ | Any linear | LoRA repair | ✗ | ✗ | ✗ |
| SVD-LLM V2 (Wang et al., 2025a) | Double SVD | Adaptive $R_w$ | Transformer | LoRA repair | ✗ | ✗ | ✗ |
| $A^3$ (Wong et al., 2025) | Attention activation informed SVD | Uniform | Transformer | ✗ | ✓ | ✗ | ✗ |
| ACIP (Genzel et al., 2025) | Weight-SVD + masking | Binary mask | Any linear | LoRA repair | ✗ | ✗ | ✓ |
| **FLEXRANK (ours)** | Online whitened data informed SVD | Pareto optimal | Any linear | Distillation | ✗ | ✓ | ✓ |

# B. Proofs

We first recall here the necessary notation. Consider a matrix $M^\star \in \mathbb{R}^{m \times n}$, parameterized as $M = UV^\top$ with $U \in \mathbb{R}^{m \times k}$, $V \in \mathbb{R}^{n \times k}$, and $k = \min(m, n)$. Assume it has singular value decomposition $M^\star = P\Sigma Q^\top$ satisfying:

$$\Sigma = \operatorname{diag}(\sigma_1, \ldots, \sigma_k), \qquad\qquad \sigma_i > \sigma_{i+1} > 0 \quad \forall i < k, \qquad (13)$$

where $P \in \mathbb{R}^{m \times k}$ and $Q \in \mathbb{R}^{n \times k}$ have orthonormal columns. We call $A_r$ the $r$-truncation of $M^\star$.

## B.1. Assumptions

**Assumption B.1.** Let $(U_0, V_0)$ be a random initialization for $(U, V)$. Then assume $(U_0, V_0)$ has a density w.r.t. the Lebesque measure, *i.e.*, $(U_0, V_0)$ is a continuous random variable which has a probability density function (PDF).

**Assumption B.2.** Let $S_r \subseteq [k], |S_r| = r$ the set of $r$ nonzero columns of a matrix and call $\Pi_{S_r}$ the diagonal projector onto coordinates in $S_r$, *i.e.* $(\Pi_{S_r})_{jj} = 1$ iff $j \in S_r$, otherwise $(\Pi_{S_r})_{jj} = 0$. Then assume gradient descent (GD) converges to a global minimizer for the problem $\arg\min_{U,V} \|U\Pi_{S_r} V^\top - A_r\|_F^2, \quad \forall r \in \{1, \ldots, k\}$.

## B.2. Auxiliary Lemmas

**Lemma B.3** (Objective equivalence without empty mask in ASL). *Let $\mathcal{L}_1(\cdot)$ and $\mathcal{L}_2(\cdot)$ be defined as:*

$$\mathcal{L}_1(U, V) := \frac{1}{2^k - 1} \sum_{\substack{S \subseteq [k] \\ S \neq \emptyset}} \|U\Pi_S V^\top - M^\star\|_F^2, \qquad \mathcal{L}_2(U, V) := \frac{1}{2^k} \sum_{S \subseteq [k]} \|U\Pi_S V^\top - M^\star\|_F^2.$$

*Then $\mathcal{L}_1$ and $\mathcal{L}_2$ have the same set of minimizers, as the following holds:*

$$\mathcal{L}_1(U, V) = \frac{2^k}{2^k - 1} \mathcal{L}_2(U, V) - \frac{1}{2^k - 1} \|M^\star\|_F^2.$$

*Proof.* The only additional term in $\mathcal{L}_2$ is the empty mask $S = \emptyset$, for which $U\Pi_S V^\top = 0$ and $\|U\Pi_S V^\top - M^\star\|_F^2 = \|M^\star\|_F^2$. Rearranging the terms yields the identity. $\square$

**Lemma B.4** (Rank-dropout objective expansion). *Let $z = (z_1, \ldots, z_k)$ have i.i.d. entries $z_j \sim \operatorname{Bernoulli}(1/2)$, and let $\Pi_z := \operatorname{diag}(z)$. Write $U = [u_1, \ldots, u_k]$ and $V = [v_1, \ldots, v_k]$. Then:*

$$\mathbb{E}_z \|U\Pi_z V^\top - M^\star\|_F^2 = \frac{1}{4} \|UV^\top - 2M^\star\|_F^2 + \frac{1}{4} \sum_{j=1}^k \|u_j\|_2^2 \|v_j\|_2^2. \qquad (14)$$

*Proof.* Let $W(z) := U\Pi_z V^\top$. Expanding the square leads to:

$$\|W(z) - M^\star\|_F^2 = \|W(z)\|_F^2 + \|M^\star\|_F^2 - 2\langle W(z), M^\star \rangle. \qquad (15)$$

Since $\mathbb{E}[\Pi_z] = \frac{1}{2}\mathbf{I}_k$, we have that:

$$\mathbb{E}_z[W(z)] = \tfrac{1}{2} UV^\top, \qquad \mathbb{E}_z \langle W(z), M^\star \rangle = \tfrac{1}{2} \langle UV^\top, M^\star \rangle. \qquad (16)$$

To evaluate the quadratic term, note that $W(z) = \sum_{j=1}^k z_j u_j v_j^\top$. Therefore,

$$\begin{aligned}
\|W(z)\|_F^2 = \left\langle \sum_{i=1}^k z_i u_i v_i^\top, \sum_{j=1}^k z_j u_j v_j^\top \right\rangle &= \sum_{i=1}^k \sum_{j=1}^k z_i z_j \langle u_i v_i^\top, u_j v_j^\top \rangle \\
&= \sum_{i=1}^k \sum_{j=1}^k z_i z_j \operatorname{tr}(v_i u_i^\top u_j v_j^\top) \\
&= \sum_{i=1}^k \sum_{j=1}^k z_i z_j (u_i^\top u_j)(v_i^\top v_j), \qquad (17)
\end{aligned}$$

where we used $\langle A, B \rangle = \mathrm{tr}(A^\top B)$ and the cyclicity of the trace. Using $\mathbb{E}[z_i^2] = \frac{1}{2}$ and $\mathbb{E}[z_i z_j] = \frac{1}{4}$ for $i \neq j$, we obtain:

$$\mathbb{E}_z \|W(z)\|_F^2 = \frac{1}{4} \|UV^\top\|_F^2 + \frac{1}{4} \sum_{j=1}^{k} \|u_j\|_2^2 \|v_j\|_2^2. \tag{18}$$

Substituting (16) and (18) into (15) yields:

$$\mathbb{E}_z \|W(z) - M^\star\|_F^2 = \frac{1}{4} \|UV^\top\|_F^2 + \|M^\star\|_F^2 - \langle UV^\top, M^\star \rangle + \frac{1}{4} \sum_{j=1}^{k} \|u_j\|_2^2 \|v_j\|_2^2$$

$$= \frac{1}{4} \|UV^\top - 2M^\star\|_F^2 + \frac{1}{4} \sum_{j=1}^{k} \|u_j\|_2^2 \|v_j\|_2^2$$

$\square$

**Lemma B.5** (Balanced factorization penalty). *Fix $W \in \mathbb{R}^{m \times n}$ with $\mathrm{rank}(W) \leq k$. Define the balanced factorization penalty*

$$\mathcal{F}_k(W) := \min_{\substack{U \in \mathbb{R}^{m \times k}, V \in \mathbb{R}^{n \times k} \\ UV^\top = W}} \sum_{j=1}^{k} \|u_j\|_2^2 \|v_j\|_2^2, \tag{19}$$

*which measures the minimal columnwise energy required to represent $W$ using $k$ rank-one components. Then:*

$$\mathcal{F}_k(W) = \frac{1}{k} \|W\|_\star^2. \tag{20}$$

*Moreover, any minimizer $(U, V)$ satisfies:*

$$\|u_j\|_2 \|v_j\|_2 = \frac{1}{k} \|W\|_\star \quad \text{for all } j \in [k], \qquad \sum_{j=1}^{k} \|u_j\|_2 \|v_j\|_2 = \|W\|_\star. \tag{21}$$

*Proof.* Let $W = \sum_{j=1}^{k} u_j v_j^\top$ be any feasible factorization and define

$$a_j := \|u_j\|_2 \|v_j\|_2 = \|u_j v_j^\top\|_\star.$$

By the triangle inequality for the nuclear norm, it holds that:

$$\|W\|_\star = \left\| \sum_{j=1}^{k} u_j v_j^\top \right\|_\star \leq \sum_{j=1}^{k} a_j. \tag{22}$$

Applying the Cauchy–Schwarz inequality yields:

$$\sum_{j=1}^{k} a_j^2 \geq \frac{1}{k} \left( \sum_{j=1}^{k} a_j \right)^2 \geq \frac{1}{k} \|W\|_\star^2. \tag{23}$$

Since $\sum_j a_j^2 = \sum_j \|u_j\|_2^2 \|v_j\|_2^2$, this shows:

$$\mathcal{F}_k(W) \geq \frac{1}{k} \|W\|_\star^2. \tag{24}$$

To show that the bound is tight, let $W = P \, \mathrm{diag}(s_1, \ldots, s_k) \, Q^\top$ be an SVD of $W$, padded with zeros if necessary, so that $\|W\|_\star = \sum_{i=1}^{k} s_i$. By the Schur–Horn theorem, there exists an orthogonal matrix $R \in \mathbb{R}^{k \times k}$ such that the diagonal entries of $R^\top \mathrm{diag}(s) R$ are all equal to:

$$\bar{s} := \frac{1}{k} \sum_{i=1}^{k} s_i = \frac{1}{k} \|W\|_\star.$$

Intuitively, this rotation redistributes the singular values uniformly across the $k$ columns. Define:

$$U := P \operatorname{diag}(\sqrt{s}) \, R, \qquad V := Q \operatorname{diag}(\sqrt{s}) \, R.$$

Then $UV^\top = W$ and, for each $j \in [k]$,

$$\|u_j\|_2^2 = (U^\top U)_{jj} = (R^\top \operatorname{diag}(s) R)_{jj} = \bar{s}, \qquad \|v_j\|_2^2 = \bar{s}.$$

Consequently:

$$\sum_{j=1}^{k} \|u_j\|_2^2 \, \|v_j\|_2^2 = k \, \bar{s}^2 = \frac{1}{k} \, \|W\|_\star^2,$$

which proves $\mathcal{F}_k(W) \leq \frac{1}{k} \|W\|_\star^2$. Finally, equality in (23) requires $a_1 = \cdots = a_k$, while equality in (22) forces additivity, *i.e.*, $\sum_{j=1}^{k} a_j = \|W\|_\star$. Recalling $a_j = \|u_j\|_2 \|v_j\|_2$ yields (21). $\qquad\square$

**Lemma B.6** (Spectral form of the minimizer of ASL objective). *Let $M^\star = P \Sigma Q^\top$ be the singular value decomposition of $M^\star$, with $\Sigma = \operatorname{diag}(\sigma_1, \ldots, \sigma_k)$ and $\sigma_i \geq 0$. Consider the objective*

$$\Phi(W) = \frac{1}{4} \|W - 2M^\star\|_F^2 + \frac{1}{4k} \|W\|_\star^2. \tag{25}$$

*The function $\Phi$ admits a unique global minimizer $W^\star$. Moreover, there exists a minimizer whose left and right singular subspaces coincide with those of $M^\star$, and such a minimizer can be written as:*

$$W^\star = P \operatorname{diag}(w_1, \ldots, w_k) \, Q^\top, \tag{26}$$

*where the singular values $w_i \geq 0$ are uniquely determined by:*

$$w_i = \max(0, \, 2\sigma_i - \lambda), \qquad i = 1, \ldots, k, \tag{27}$$

*with $\lambda$ satisfying the consistency condition*

$$\lambda = \frac{1}{k} \sum_{j=1}^{k} w_j. \tag{28}$$

*Proof.* Expanding the Frobenius norm in (25) yields

$$\Phi(W) = \frac{1}{4} \|W\|_F^2 - \langle W, M^\star \rangle + \frac{1}{4k} \|W\|_\star^2 + \|M^\star\|_F^2. \tag{29}$$

The terms $\|W\|_F^2$ and $\|W\|_\star$ depend only on the singular values of $W$. Let $W = \tilde{P} \operatorname{diag}(w) \tilde{Q}^\top$ be an arbitrary SVD. For fixed $w$, the inner product $\langle W, M^\star \rangle$ is maximized, by the von Neumann trace inequality, when the singular subspaces of $W$ align with those of $M^\star$. Since $\Phi(W)$ contains $-\langle W, M^\star \rangle$, for any fixed singular values the objective is minimized when the singular subspaces of $W$ align with those of $M^\star$. Restricting to this form yields the reduced problem:

$$\min_{w \geq 0} \phi(w) := \frac{1}{4} \sum_{i=1}^{k} (w_i - 2\sigma_i)^2 + \frac{1}{4k} \left( \sum_{j=1}^{k} w_j \right)^2. \tag{30}$$

The function $\phi$ is strictly convex on $\mathbb{R}^k$ and therefore admits a unique minimizer. Introduce Lagrange multipliers $\mu_i \geq 0$ associated with the constraints $w_i \geq 0$. The Lagrangian is:

$$\mathcal{L}(w, \mu) = \frac{1}{4} \sum_{i=1}^{k} (w_i - 2\sigma_i)^2 + \frac{1}{4k} \left( \sum_{j=1}^{k} w_j \right)^2 - \sum_{i=1}^{k} \mu_i w_i. \tag{31}$$

Stationarity with respect to $w_i$ yields:

$$\frac{1}{2}(w_i - 2\sigma_i) + \frac{1}{2k} \sum_{j=1}^{k} w_j - \mu_i = 0. \tag{32}$$

Defining $\lambda := \frac{1}{k} \sum_{j=1}^{k} w_j$, and using feasibility $w_i, \mu_i \geq 0$ together with complementary slackness $\mu_i w_i = 0$, the KKT conditions are equivalent to:

$$w_i = \max(0, 2\sigma_i - \lambda), \qquad i = 1, \ldots, k. \tag{33}$$

Summing (33) over $i$ enforces the consistency condition (28), completing the proof. $\qquad\square$

**Theorem B.7** (Spectral interference and impossibility of perfect recovery). *Let $M^\star = P\Sigma Q^\top$ have singular values $\sigma_1 > \sigma_2 > \cdots > \sigma_k > 0$, and let $W^\star$ be the unique minimizer of $\Phi(W)$ as defined in Lemma B.6. Then $W^\star = M^\star$ if and only if*

$$\sigma_1 = \sigma_2 = \cdots = \sigma_k. \tag{34}$$

*Consequently, under the standing assumption $\sigma_1 > \cdots > \sigma_k$, every global minimizer $(U, V)$ of ASL satisfies:*

$$\|UV^\top - M^\star\|_F^2 = \|W^\star - M^\star\|_F^2 > 0. \tag{35}$$

*Proof.* By Lemma B.6, the objective $\Phi$ admits a unique minimizer $W^\star$, whose singular subspaces align with those of $M^\star$, and whose singular values $w_1, \ldots, w_k$ satisfy

$$w_i = \max(0, 2\sigma_i - \lambda), \qquad \lambda = \frac{1}{k} \sum_{j=1}^{k} w_j, \quad i = 1, \ldots, k. \tag{36}$$

Suppose that $W^\star = M^\star$. Then $w_i = \sigma_i$ for all $i$, and (36) implies

$$\sigma_i = 2\sigma_i - \lambda, \qquad i = 1, \ldots, k. \tag{37}$$

Hence $\sigma_i = \lambda$ for all $i$, which yields $\sigma_1 = \cdots = \sigma_k$. Conversely, if $\sigma_1 = \cdots = \sigma_k$, then setting $w_i = \sigma_i$ satisfies (36). By uniqueness of the minimizer of $\Phi$, this implies $W^\star = M^\star$.

Under the standing assumption $\sigma_1 > \cdots > \sigma_k$, the equality $W^\star = M^\star$ is therefore impossible. Since $W^\star$ is unique, it follows that: $\|W^\star - M^\star\|_F^2 > 0$, and consequently

$$\|UV^\top - M^\star\|_F^2 = \|W^\star - M^\star\|_F^2 > 0$$

for every global minimizer $(U, V)$ of the objective in Eq. (11). $\qquad\square$

## B.3. Proofs of Main Theorems

B.3.1. PROOF OF THM. 4.1: (PTS HAS NONGENERIC RANK REDUCTION)

We establish the theorem by characterizing the algebraic structure of the set of global minimizers $\mathcal{M}$ and evaluating the measure of the subset permitting submodel recovery.

**Case 1: Perfect recovery of the full solution ($r = k$).** By Assumption B.2, any global minimizer satisfies the zero-loss condition $UV^\top = M^\star$. Since the target $M^\star$ is rank-$k$, it follows that $M^\star = A_k$. Substituting this into the definition of the submodel gap yields:

$$\mathcal{E}(U, V, k) = \|UV^\top - A_k\|_F^2 = \|M^\star - M^\star\|_F^2 = 0. \tag{38}$$

**Case 2: Nongenericity of the optimal solution for $r < k$.** Any global minimizer $(U, V) \in \mathcal{M}$ can be parameterized via a gauge transformation $R \in \mathrm{GL}(k)$ relative to the SVD factors $M^\star = P\Sigma Q^\top$:

$$U = P\Sigma^{1/2}R, \quad V = Q\Sigma^{1/2}R^{-\top}. \tag{39}$$

For a fixed $r < k$, the condition $\mathcal{E}(U, V, r) = 0$ implies there exists a subset $S_r \subseteq [k]$ with $|S_r| = r$ such that $U\Pi_{S_r}V^\top = A_r$. Substituting the parameterization from (39), we obtain:

$$P\Sigma^{1/2}R\Pi_{S_r}R^{-1}\Sigma^{1/2}Q^\top = P\Sigma^{1/2}\Pi_{[r]}\Sigma^{1/2}Q^\top$$
$$\iff R\Pi_{S_r} = \Pi_{[r]}R. \tag{40}$$

Let $C_\pi$ be the permutation matrix mapping the indices in $S_r$ to the first $r$ integers. The commutation relation (40) forces $RC_\pi$ to possess a specific block-diagonal structure:

$$RC_\pi = \begin{pmatrix} R_{11} & 0 \\ 0 & R_{22} \end{pmatrix}, \quad R_{11} \in \mathrm{GL}(r), \quad R_{22} \in \mathrm{GL}(k-r). \tag{41}$$

Let $\mathcal{H}_{S_r} \subset \mathrm{GL}(k)$ be the set of matrices satisfying (41). Since $\mathrm{GL}(k)$ is an open subset of the vector space $\mathbb{R}^{k^2}$, we can evaluate the measure of $\mathcal{H}_{S_r}$ by its codimension:

$$\begin{aligned} \mathrm{codim}(\mathcal{H}_{S_r}) &= \dim(\mathbb{R}^{k^2}) - \dim(\mathcal{H}_{S_r}) \\ &= k^2 - \left(r^2 + (k-r)^2\right) = 2r(k-r). \end{aligned} \tag{42}$$

For any $1 \leq r < k$, the codimension is at least 2. As a proper lower-dimensional subset of $\mathbb{R}^{k^2}$, $\mathcal{H}_{S_r}$ has Lebesgue measure zero. Under Assumption B.1, the gauge $R$ is determined by a random initialization with an absolutely continuous distribution, which implies $\mathbb{P}[R \in \mathcal{H}_{S_r}] = 0$ for any specific subset $S_r$.

To show this holds for all possible submodels, we apply the union bound over the finite collection of all subsets $\mathcal{S}_r = \{S \subseteq [k] : |S| = r\}$:

$$\mathbb{P}[\mathcal{E}(U,V,r) = 0] = \mathbb{P}\left[\bigcup_{S \in \mathcal{S}_r} R \in \mathcal{H}_S\right] \leq \sum_{S \in \mathcal{S}_r} \mathbb{P}[R \in \mathcal{H}_S] = 0. \tag{43}$$

Finally, applying the union bound over all truncation levels $r \in \{1, \ldots, k-1\}$ yields:

$$\mathbb{P}\left[\exists r < k : \mathcal{E}(U,V,r) = 0\right] \leq \sum_{r=1}^{k-1} \mathbb{P}[\mathcal{E}(U,V,r) = 0] = 0. \tag{44}$$

Thus, for a global minimizer $(U,V)$ reached via GD, the condition $\mathcal{E}(U,V,r) > 0$ for all $r < k$ holds with probability 1.

### B.3.2. PROOF OF THM. 4.2: (ASL HAS STRICTLY POSITIVE SUBMODEL GAP)

Consider the objective in Eq. (11), by Lemma B.3 its optimal solution coincide with the one of the objective function over all the masks (*i.e.* including the empty mask). In particular, by Lemmas B.4 and B.5 minimizing the ASL objective is equivalent to minimize:

$$\Phi(W) = \frac{1}{4}\|W - 2M^\star\|_F^2 + \frac{1}{4k}\|W\|_\star^2. \tag{45}$$

Let $W^\star = P\mathrm{diag}(\mathbf{w})Q^\top$ the SVD decomposition of the optimal solution given by Thm. B.7, where $\mathbf{w} \in \mathbb{R}^k$ are the eigenvalues and $P, Q$ are orthonormal matrices. Define the dual test matrix $G := PQ^\top$. By orthonormality of $P$ and $Q$,

$$\|G\|_F = \sqrt{k}, \qquad \|G\|_{\mathrm{op}} = 1. \tag{46}$$

By Lemma B.6, $W^\star$ shares singular subspaces with $M^\star$. Consequently, for the rank-$r$ truncation $A_r$ of $M^\star$,

$$\langle G, A_r \rangle = \sum_{i=1}^r \sigma_i, \qquad \langle G, W^\star \rangle = \sum_{i=1}^k w_i = \|W^\star\|_\star. \tag{47}$$

Let $W^\star = \sum_{j=1}^k u_j v_j^\top$ be any optimal factorization. By Lemma B.6,

$$\|u_j\|_2 \|v_j\|_2 = \frac{1}{k}\|W^\star\|_\star = \lambda, \qquad j = 1, \ldots, k. \tag{48}$$

Using duality between operator and nuclear norms together with (46):

$$\langle G, u_j v_j^\top \rangle \leq \|G\|_{\mathrm{op}}\|u_j v_j^\top\|_\star = \lambda. \tag{49}$$

Since $\sum_{j=1}^k \langle G, u_j v_j^\top \rangle = \langle G, W^\star \rangle = k\lambda$, each inequality in (49) must be tight. Therefore:

$$\langle G, U\Pi_S V^\top \rangle = r\lambda \tag{50}$$

for every subset $S \subset [k]$ with $|S| = r$. By the Cauchy-Schwarz inequality, the Frobenius distance is bounded by its projection onto the direction of $G$:

$$\|U\Pi_S V^\top - A_r\|_F \geq \frac{|\langle G, U\Pi_S V^\top - A_r\rangle|}{\|G\|_F}$$

$$= \frac{|r\lambda - \sum_{i=1}^{r} \sigma_i|}{\sqrt{k}}. \tag{51}$$

Squaring yields the stated lower bound on $\mathcal{E}(U, V, r)$. To prove that $\mathcal{E}(U, V, r) > 0$ generically, define for each $r = 1, \ldots, k$ the linear functional $f_r$ on the spectrum $\boldsymbol{\sigma} = (\sigma_1, \ldots, \sigma_k) \in \mathbb{R}^k$ by

$$f_r(\boldsymbol{\sigma}) := r\lambda - \sum_{i=1}^{r} \sigma_i = \left(\frac{r-k}{k}\right)\sum_{i=1}^{r} \sigma_i + \frac{r}{k}\sum_{i=r+1}^{k} \sigma_i. \tag{52}$$

For each $r$, the zero set:

$$\mathcal{Z}_r := \{\boldsymbol{\sigma} \in \mathbb{R}^k : f_r(\boldsymbol{\sigma}) = 0\} \tag{53}$$

is a proper linear subspace of codimension one. By basic properties of Lebesgue measure, each $\mathcal{Z}_r$ has measure zero. Finally, applying the union bound over all submodel sizes $r = 1, \ldots, k$, we obtain:

$$\mathbb{P}\left[\bigcup_{r=1}^{k} \mathcal{Z}_r\right] \leq \sum_{r=1}^{k} \mathbb{P}[\boldsymbol{\sigma} \in \mathcal{Z}_r] = 0. \tag{54}$$

Thus, for any absolutely continuous distribution on $\mathbb{R}^k$, the gap condition $\mathcal{E}(U, V, r) > 0$ holds simultaneously for all $r = 1, \ldots, k$ with probability 1.

**Corollary B.8.** *Training all possible submodels can lead to suboptimal solution $\hat{U}, \hat{V}$, i.e.:*

$$(\hat{U}, \hat{V}) \notin \mathcal{M} := \{(U, V) \in \mathbb{R}^{m \times k} \times \mathbb{R}^{n \times k} : UV^\top = M^\star\}.$$

*Proof sketch.* For ease of exposition, we consider the smaller setup, where $A \in \mathbb{R}^{2 \times 2}$. If we train all the models together, the optimization problem is

$$\min_{U,V} \frac{1}{3}\Big(\|U_{\{1\}}V_{\{1\}}^\top - A\|_F^2 + \|U_{\{2\}}V_{\{2\}}^\top - A\|_F^2$$

$$+ \|U_{\{1,2\}}V_{\{1,2\}}^\top - A\|_F^2\Big). \tag{55}$$

If we recover the optimal Pareto front, it has to be the case that $U_{\{1\}}V_{\{1\}}^\top = A_1$ and $U_{\{2\}}V_{\{2\}}^\top = A_2 - A_1$ (indices could be flipped). Plugging this back to (55), the final objective value is equal to $\sigma_2^2 + \sigma_1^2 + 0$, where $\sigma_1 \geq \sigma_2$ are eigenvalues of A. On the other hand, if we have $U_{\{1\}}V_{\{1\}}^\top = U_{\{2\}}V_{\{2\}}^\top = cA_1$, then the objective value is equal to $3\sigma_2^2 + (2(1-c)^2 + (1-2c)^2)\sigma_1^2$. If $c = 2/3$, then the objective is equal to $3\sigma_2^2 + \sigma_1^2/3$. Therefore, if $\sigma_1^2 > 3\sigma_2^2$, the second solution is better than the first one, and we do not recover the optimal Pareto front, which concludes the proof. $\square$

### B.3.3. PROOF OF THM. 4.3: (NSL PRESERVES NESTED MINIMIZERS)

Let $(U, V) \in \mathcal{M}_{\text{NSL}}$. By the Eckart–Young–Mirsky theorem, for each $r \in \{1, \ldots, k\}$, the $r$-rank matrix $U\Pi_{[r]}$ satisfies the lower bound $\|U\Pi_{[r]}V^\top - M^\star\|_F^2 \geq \|A_r - M^\star\|_F^2$. Since the SVD factors of $M^\star$ achieve all $k$ lower bounds simultaneously, any global minimizer $(U, V)$ must satisfy:

$$U\Pi_{[r]}V^\top = A_r \quad \text{for all } r \in \{1, \ldots, k\}. \tag{56}$$

We claim that the NSL objective enforces the structural constrains in (56). We proceed by induction.

**Base case ($r = 1$).** For $r = 1$, the condition (56) requires:

$$U\Pi_{[1]}V^\top = A_1 \implies u_1 v_1^\top = \sigma_1 p_1 q_1^\top. \tag{57}$$

**Inductive step.** Assume that for some $r \in \{2, \ldots, k\}$, the inductive hypothesis $U\Pi_{[r-1]}V^\top = A_{r-1}$ holds. By the global optimality condition (56), the $r$-th submodel must satisfy:

$$U\Pi_{[r]}V^\top = A_r. \tag{58}$$

Recalling the recursive definition of the submodels $U\Pi_{[r]}V^\top = U\Pi_{[r-1]}V^\top + u_r v_r^\top$, we substitute the inductive hypothesis into (58):

$$
\begin{aligned}
A_{r-1} + u_r v_r^\top &= A_r \\
\implies u_r v_r^\top &= A_r - A_{r-1} \\
&= \sigma_r p_r q_r^\top
\end{aligned} \tag{59}
$$

where the final equality follows from the recursive structure of the truncated SVD. Then by induction we have that the relation in (57) holds for any $r$. Consequently, it holds that $\mathcal{E}(U, V, r) = 0 \; \forall r \in \{1, \ldots, k\}$, which concludes the proof.

# C. Additional Details

## C.1. Layer Decomposition

Directly solving (3) for a large sample of activations is memory-prohibitive, as storing $\mathbf{X}_l$ scales with $\mathcal{O}(N \cdot n_l)$. We instead utilize an efficient variant based on the second moment of the activations. Observing that $\|A\mathbf{X}\|_F^2 = \mathrm{Tr}(A\mathbf{X}\mathbf{X}^\top A^\top)$, we rewrite the objective as

$$
\begin{aligned}
\|(\theta_l - U_l V_l^\top)\mathbf{X}_l\|_F^2 &= \mathrm{Tr}\left[\Delta\theta_l \Sigma_l \Delta\theta_l^\top\right] \\
&= \|(\theta_l - U_l V_l^\top)\Sigma_l^{1/2}\|_F^2
\end{aligned} \tag{60}
$$

where $\Delta\theta_l = (\Delta\theta_l - U_l V_l^\top)$ and $\Sigma_l = \mathbf{X}_l \mathbf{X}_l^\top \in \mathbb{R}^{n_l \times n_l}$ is the unnormalized covariance matrix. This enables a two-stage initialization

**1. Online Covariance Estimation:** We batch-accumulate $\Sigma_l = \sum_j \mathbf{x}_{l,j}\mathbf{x}_{l,j}^\top$ by running batches of activations through the model. Note that the memory complexity is now independent of $N$ and scales as $\mathcal{O}(n_l^2)$.

**2. Whitened SVD:** We compute the symmetric square root $\Sigma_l^{1/2}$ via eigen-decomposition and perform SVD on the "whitened" weights $\tilde{\theta}_l = \theta_l \Sigma_l^{1/2}$, yielding $\tilde{\theta}_l = P_L \Lambda_L Q_l^\top$. To recover the factors in the original space, we observe that $\theta_l = (P_l \Lambda_l Q_l^\top)\Sigma_l^{-1/2}$. We then initialize the shared factors by symmetrically absorbing the singular values $\Lambda_l$

$$U_l \leftarrow P_l \Lambda_l^{1/2}, \quad V_l \leftarrow \Sigma_l^{-1/2} Q_l \Lambda_l^{1/2}. \tag{61}$$

This data-aware initialization aligns the rank-reduction process with the most salient directions of the feature space, providing a superior starting point for elastic training.

## C.2. Identifying Pareto Front

To solve Eq. (4) efficiently, we employ a two-step procedure:

**1. Layer Probing:** We evaluate the model's sensitivity to rank reduction at each layer independently. For each layer $l \in \{1, \ldots, L\}$ and each budget $\beta_k \in \tilde{\mathcal{B}}$, we instantiate a model where only the $l$-th layer is transformed by $\mathcal{T}_{\beta_k}$ while all other layers remain at full capacity. We record the resulting performance $\mathcal{R}_{l,k}$, constructing a sensitivity matrix $\mathbf{S} \in \mathbb{R}^{L \times K}$.

**2. Dynamic Programming Exploration:** Using the sensitivity matrix $\mathbf{S}$, we solve for the entire set of optimal configurations $\mathcal{M}$ simultaneously by framing the search as a Multi-Choice Knapsack Problem (MCKP). We employ a Dynamic Programming (DP) algorithm to find the rank assignments across layers that maximize the aggregate performance for each global threshold $\beta_k$. While DP provides an exact solution under the assumption that errors are additive across layers—a simplification of the non-linear dependencies in deep networks—our objective is not absolute optimality during probing, but rather the identification of configurations that satisfy the ranking consistency. Moreover, this algorithm is suitable for the problem in Eq. (4) since the nested constraint $\mathbf{m}_{k-1} \preceq \mathbf{m}_k$ can be enforced. The resulting procedure is summarized

---

**Algorithm 2** FLEXRANK dynamic programming (DP) rank selection (see subroutines in Algorithm 3

---

**Require:** Layer candidates $\{\mathcal{C}_\ell\}_{\ell=1}^L$, where $\mathcal{C}_\ell = \{(s_{\ell j}, e_{\ell j})\}_j$ contains rank savings and reconstruction errors
**Ensure:** Componentwise-nested Pareto configurations $\mathcal{P}$
1: $\mathcal{F} \leftarrow \{(0,0)\}$                                                             ▷ frontier: total saving, total error
2: $\mathcal{B} \leftarrow []$                                                                      ▷ backpointers
3: **for** $\ell = 1, \dots, L$ **do**
4:     $\mathcal{A} \leftarrow$ EXPANDLAYER$(\mathcal{F}, \mathcal{C}_\ell)$
5:     $\mathcal{A} \leftarrow$ KEEPMINERRORPERSAVING$(\mathcal{A})$
6:     $(\mathcal{F}, B_\ell) \leftarrow$ PARETOPRUNE$(\mathcal{A})$
7:     append $B_\ell$ to $\mathcal{B}$
8: **end for**
9: $\mathcal{P} \leftarrow$ BACKTRACK$(\mathcal{F}, \mathcal{B})$
10: $\mathcal{P} \leftarrow$ PARETOFILTER$(\mathcal{P})$
11: $\mathcal{P} \leftarrow$ NESTEDCHAIN$(\mathcal{P})$
12: **return** $\mathcal{P}$

---

in Algorithms 2 and 3: the DP expands all layer-wise rank-drop candidates, keeps only the minimum-error state for each total saving, Pareto-prunes dominated states, reconstructs configurations through backpointers, and finally retains a componentwise-nested chain.

**Complexity Analysis.** Let $C_{\text{eval}}$ denote the maximum cost of a single model evaluation. While a brute-force search requires $\mathcal{O}(K^L \cdot C_{\text{eval}})$, our approach reduces the probing cost to $\mathcal{O}(L \cdot K \cdot C_{\text{eval}})$. The subsequent DP exploration operates on the pre-computed matrix $\mathbf{S}$ with complexity $\mathcal{O}(L \cdot K)$, making the total cost of identifying the entire Pareto front linear in both the number of layers and the budget levels.

## C.3. Analysis of ranking-preservation assumption

In this section we analyze the assumption behind the use of our dynamic programming (DP) algorithm to solve the optimal rank-assignment problem over a pretrained model's layers. The DP assumption should not be interpreted as an exact theorem/solution: the role of the additive probe is to make the otherwise combinatorial rank-allocation problem tractable. We begin by noting that, while the DP algorithm uses an additive probe to estimate the loss induced by rank reduction, what is needed to preserve optimal candidates is not the exact additivity, but that the probe induces a sufficiently reliable ordering over candidate solutions, which is substantially weaker. This is the intended read of the method in this paper. The question, therefore, becomes whether this *"ranking-preservation"* is reasonable in practice. Our view is that, while it cannot be tractably shown in full generality for arbitrary deep nonlinear networks, it is reasonable in the regimes we study.

**Notation.** Call $m = (m_1, \dots, m_L)$ any submodel induced by a rank-selection over $L$ layers, and $s(m) = (s_{m_1}, \dots, s_{m_L})$, where $s_{m_l}$ is the model's sensitivity to rank reduction at each layer independently. Define $A(m) := \sum_{l=1}^L s_{m_l}$ the additive probe used by DP, and $F(m)$ the true joint probing loss, obtained by directly testing model . The experiment is conducted on MNIST on a search space of $10^4$ submodels (see details in App. D.1).

**Metrics.** We consider the following metrics:

- $\rho \in [-1, 1]$: it is the Spearman's correlation between $A(m)$ and $F(m)$; the closer to 1 the stronger the ranking agreement
- $\nu \in [0, 1]$: it is the pairwise violation rate, measuring the ranking-inversions; a value closer to 0 indicates fewer inversions.
- $p \in [0, 1]$: it is the fraction of cases in which the optimal model found by DP is also optimal w.r.t. $F(m)$, higher is better.
- $r_{avg}, r_{max}$, relative to the minimum evaluation loss: it is the margin between $A(m)$ and $F(m)$ when DP does not choose the best model w.r.t. $F(m)$, lower is better.

**Discussion.** Results presented in Fig. 9 show **very high global-level rank-agreement (Fig. A), high DP success rate (Fig. B) and low regret (Fig. C)**. In Fig. A, each point is one submodel, and the plot shows the submodel's position in the global ranking, *i.e.*, a point $(x, y)$ reads as: *"this submodel is in the top $x\%$ for $A(m)$ and in the top $y\%$ for $F(m)$"*. Values near the diagonal mean that $A(m)$ and $F(m)$ give that submodel a similar global ranking. The Spearman coefficient $\rho = 0.991$ and violation rate $\nu = 3.7\%$ (tab. D) indicate very strong global monotone agreement and very low global ranking error.

In Fig. B we show that DP matches the true exact-budget winner on $p = 94.1\%$ of cases over all the front of best submodels;

**Algorithm 3** Subroutines for Algorithm 2

| |
|---|

1: **function** EXPANDLAYER($\mathcal{F}, \mathcal{C}$)
2: $\mathcal{A} \leftarrow \emptyset$
3: **for all** $i = 1, \ldots, |\mathcal{F}|$ **do**
4:    let $(S_i, E_i)$ be the $i$-th state in $\mathcal{F}$
5:    **for all** $(s, e) \in \mathcal{C}$ **do**
6:       add $(S_i + s, E_i + e, i, s)$ to $\mathcal{A}$
7:    **end for**
8:    add $(S_i, E_i, i, 0)$ to $\mathcal{A}$        $\triangleright$ no saving
9: **end for**
10: **return** $\mathcal{A}$
11: **end function**

1: **function** KEEPMINERRORPERSAVING($\mathcal{A}$)
2: $\mathcal{A}' \leftarrow \emptyset$
3: **for all** unique total savings $S$ in $\mathcal{A}$ **do**
4:    add to $\mathcal{A}'$ the candidate with saving $S$ and minimum error
5: **end for**
6: **return** $\mathcal{A}'$
7: **end function**

1: **function** PARETOPRUNE($\mathcal{A}$)
2: sort $\mathcal{A}$ by increasing total saving
3: $\mathcal{F} \leftarrow [\,], B \leftarrow [\,], E_{\text{best}} \leftarrow \infty$
4: **for all** candidate $(S, E, i, s)$ in $\mathcal{A}$ scanned from largest to smallest $S$ **do**
5:    **if** $E < E_{\text{best}}$ **then**
6:       prepend $(S, E)$ to $\mathcal{F}$
7:       prepend $(i, s)$ to $B$
8:       $E_{\text{best}} \leftarrow E$
9:    **end if**
10: **end for**
11: **return** $\mathcal{F}, B$
12: **end function**

1: **function** BACKTRACK($\mathcal{F}, \mathcal{B}$)
2: $\mathcal{P} \leftarrow \emptyset$
3: **for all** final state index $i = 1, \ldots, |\mathcal{F}|$ **do**
4:    let $(S, E)$ be state $i$ in $\mathcal{F}$ and set $h \leftarrow i$
5:    initialize $d \leftarrow (0, \ldots, 0) \in \mathbb{Z}^L$
6:    **for** $\ell = L, \ldots, 1$ **do**
7:       let $(h', s)$ be the $h$-th backpointer in $B_\ell$
8:       $d_\ell \leftarrow s$ and $h \leftarrow h'$
9:    **end for**
10:    add $(E, d)$ to $\mathcal{P}$
11: **end for**
12: **return** $\mathcal{P}$
13: **end function**

1: **function** PARETOFILTER($\mathcal{P}$)
2: sort $\mathcal{P}$ by increasing $\sum_\ell d_\ell$
3: $\mathcal{P}' \leftarrow [\,], E_{\text{best}} \leftarrow \infty$
4: **for all** $(E, d) \in \mathcal{P}$ scanned from largest to smallest $\sum_\ell d_\ell$ **do**
5:    **if** $E < E_{\text{best}}$ **then**
6:       prepend $(E, d)$ to $\mathcal{P}'$
7:       $E_{\text{best}} \leftarrow E$
8:    **end if**
9: **end for**
10: **return** $\mathcal{P}'$
11: **end function**

1: **function** NESTEDCHAIN($\mathcal{P}$)
2: $\mathcal{P}' \leftarrow [\,]$
3: **for all** $(E, d) \in \mathcal{P}$ sorted by increasing $\sum_\ell d_\ell$ **do**
4:    **if** $\mathcal{P}' = \emptyset$ or $d \geq d_{\text{last}}$ componentwise **then**
5:       append $(E, d)$ to $\mathcal{P}'$
6:       $d_{\text{last}} \leftarrow d$
7:    **end if**
8: **end for**
9: **return** $\mathcal{P}'$
10: **end function**

more importantly, when it does not return the best submodel, the regret is low. In particular, Figure C shows the empirical CDF of the resulting relative regret: for any threshold on the $x$-axis, the $y$-value gives the fraction of DP failures whose regret is at most that threshold. The fact that this curve rises quickly in the low-regret region and saturates before $12\%$ shows that most failures remain close to the true exact-budget optimum. In practice, DP often finds the same solution we would have found by exhaustive exploration, and when it does not, the solution it finds is roughly equivalent in terms of loss.

# D. Experimental Setting

## D.1. Details on controlled experiments

For the controlled experiment, we consider a simple neural network with two fully connected linear layers without any activation or bias. We generate data $\mathcal{D} \sim \mathcal{N}(0, \mathbf{I}_{10})$ and labels $\mathbf{y} = M^\star \mathbf{d} + E$, $\mathbf{d} \in \mathcal{D}$, where $M^\star \in R^{10 \times 10}$ is a randomly generated matrix whose singular values follow a power law with decay $1.2$, and $E \sim \mathcal{N}(0, 0.1)$ is an independent random noise. If we use $\ell_2$ as a loss function, the elastic training objective is exactly the same of Eq. (8), with $k = 10$ being the hidden dimension of the neural network.

## D.2. Datasets and Models

**Training.**   The core component of FLEXRANK is nested submodel training starting from a suitable initialization (*e.g.* a SVD of layer's weights). This training step involves knowledge distillation from the full model, and can be carried out using any dataset representative enough of the pretraining task. The use of a proxy dataset is in practice a requirement shared with

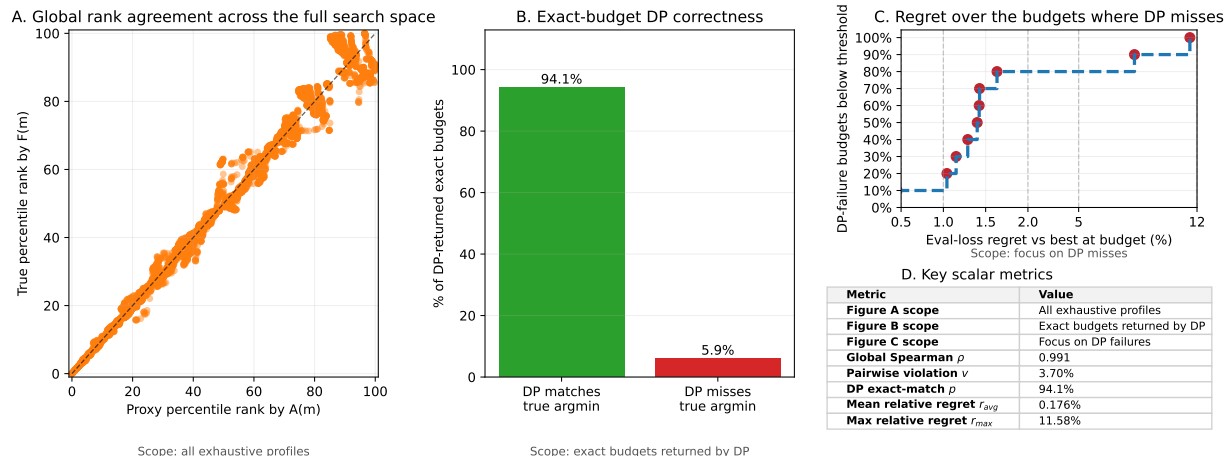

*Figure 9.* Analysis of the validity of ranking preservation assumption for DP submodel selection

other methods (Genzel et al., 2025; Cai et al., 2024; 2025). Since data quality has been proven to play an important role in language modelling, for NLP experiments we chose the FineWebEdu dataset, in particular the split composed of 10 billion tokens.

For CV experiments, we adopt the DINOv3 models family and train on ImageNet1K. More in detail, following the protocol of (Siméoni et al., 2025), we initialize the backbone with the pretrained weights publicly released and only additionally train the classification head. The resulting architecture is then treated as the input of FLEXRANK, similarly as in the NLP case.

It is important to notice that we do not finetune the backbone weights on the proxy dataset: this is the intended use of DINOv3 models on downstream tasks from the official paper, as well as their evaluation protocol. While finetuning all the weights can potentially lead to higher accuracy, we argue that preserving the backbone performance is closer to the intended use of DINOv3 models. Moreover, the use of a classification head as proxy task for knowledge distillation can be in principle exchanged with feature matching or other type of losses.

**Downstream Task Evaluation.** For NLP models, we evaluate the zero-shot performance on commonsense datasets `ARC_Challenge`, `ARC_Easy`, `HellaSwag`, `OpenBookQA`, `PIQA`, `Winogrande`, using the `lm-eval-harness` tool (Gao et al., 2024). Additionally, we evaluate the post-adaptation capabilities of FLEXRANK submodels by finetuning LoRA adapters separately for each of submodel we evaluate. This practice is simple enough to show that submodels retain sufficient knowledge from the pretraining task to be easily finetuned, and corresponds to the common scenario in which PEFT techniques are used. We follow the same experimental protocol of (Meng et al., 2024), finetuning the adapters on MetaMathQA for math domain and on Code–Feedback for coding tasks. Then, we test the 5-shot performances respectively on `MathQA` and `GSM8K`, zero-shot and on `HumanEval` and 3-shot on `MBPP`.

### D.3. Implementation Details

**Hyperparameters.** For all NLP models used in this study, we use a global batch size of 512 and a sequence length of 1024, and train for 10.000 steps, accounting for roughly half of a complete epoch (5B tokens). We use AdamW with standard parameters and learning rate $\eta = 1e - 5$ with 715 warmup steps and cosine annealing schedule.

For pretraining the classification heads of DINOv3 models, we follow the author suggestions (Siméoni et al., 2025), using SGD with momentum $\beta = 0.9$ and search the learning rate $\eta \in \{0.01, 0.02, 0.05, 0.1, 0.2, 0.5, 1, 2, 5\}$ and the weight decay $\lambda \in \{0, 1e - 5\}$, with best found values $\eta = 0.5$ and $\lambda = 0$. For all the DINOv3 models, we use a global batch size of 1024, and train for 25.000 steps, which correspond to about 20 epochs. For the training step of FLEXRANK, we use AdamW with standard parameters and learning rate $\eta = 1e - 5$ with 715 warmup steps and cosine annealing schedule.

**Reparametrizing layers into** $(U, V)$ **form.** This reparameterization is broadly applicable to the standard layers found in modern Deep Learning architectures. For **linear layers**, the shared parameters consist of the low-rank factors $\theta_l = \{U_l, V_l\}$ such that the weight matrix is $W_l = U_l V_l^\top$. For **convolutional layers** with $C_{out}$ filters, $C_{in}$ channels, and spatial dimensions

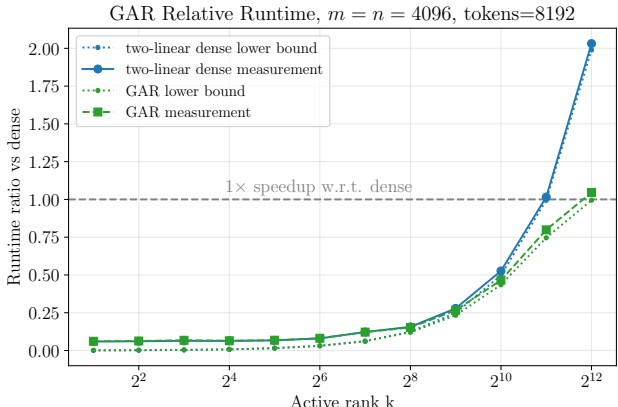

*Figure 10.* **GAR enables practical speedup following the theoretical prediction:** Relative cost of a forward pass of *(i)* the usual (blue) and the GAR factorizations (Sec. 3.5) relative to the forward of a matrix $M \in \mathbb{R}^{m \times n}$, varying the active rank $k$, for $m = n = 4096$ and an input of $8192$ tokens. All measurements have been profiled on one NVIDIA RTX5090 GPU with `torch.compile` activated.

$h \times h$, we apply the factorization to the reshaped weight matrix $M_l \in \mathbb{R}^{C_{out} \times (C_{in} \cdot h^2)}$. For **Multi-Head Attention (MHA)** modules, the implementation of $\theta_l$ follows the underlying architecture of $f$. If $f$ utilizes a fused query-key-value operator, $\theta_l$ factorizes the consolidated projection matrix; otherwise, it consists of distinct factor pairs $\{U_q, V_q\}, \{U_k, V_k\}, \{U_v, V_v\}$.

### D.4. Training and inference efficiency

**At training time.** Algorithmically, each FlexRank training step is comparable to standard distillation. However, the convergence is much faster thanks to the initialization from the pretrained model, so the required training is much lighter (*e.g.* we uniformly use 5B tokens for LM experiments, while the documented recipe the LLama3.2-1B reports a training budget of 9T tokens). However, an under-optimized implementation can lead to a higher cost. There are two implementation-level factors currently increasing latency in our experiments:

1. **Two-model for distillation.** Training requires both the teacher and student, doubling forward-pass cost and increasing memory pressure, limiting batch size and throughput. One could disaggregate model hosting and allow for asynchronous rollouts through vllm/sglang.
2. **Parameterization overhead.** The factorized layers introduce additional memory movement (*e.g.*, intermediate activations in $B@(X@A)$), making the implementation more memory-bound than dense layers when not fused. Implementing the factorized forward pass as a single fused kernel could recover the memory bottleneck.

**At inference time.** Many of the above problems do not occur at inference time, when the teacher is not used. Moreover, according to our measurements in Fig. 10, the practical scaling closely follows the one prescribed by theory: in particular, the standard factorization is up to $2\times$ as costly as a dense forward for high active rank, whereas our GAR factorization overcomes this issue, always attaining a cost reduction, even for almost full rank layers.

