# OpenReview forum: "FlexRank: Nested Low-Rank Knowledge Decomposition for Adaptive Model Deployment"
_ICML.cc/2026/Conference — ICML 2026 spotlight_

### Official Review · Reviewer_WDVJ · 2026-02-20

**Soundness:** 3
**Presentation:** 2
**Significance:** 2
**Originality:** 3
**Overall Recommendation:** 5
**Confidence:** 5

**Summary:**

The paper presents a new approach for low rank compression of large NLP and vision models. The approach starts by searching the a low rank decomposed model for configurations for a range of preselect budgets. The search is based on dynamic programming and constrained to ensure that a model at a lower budget maintains fewer parameters in every single layer than its respective, larger counterparts. Subsequently, the found configurations are used in a supernet style training proposed in the paper. Authors rigorously evaluate different options to perform that training and derive a method that yields high performance results in all configurations. Using the proposed method, authors demonstrate high performance across both NLP and CV models when compared with other approaches.

**Compliance With Llm Reviewing Policy:**

Affirmed.

**Final Justification:**

Authors have directly addressed many of my inquiries. Considering the content of all responses and the main body of the paper, I think that there are findings that are interesting in the context of rank allocation and supernet training. While the extensive cost of supernet approaches still limits significance in my opinion (comparing to distillation is nice, but not the key comparison), overall, the added information provides a good evaluation of the capabilities and improves understanding of how to construct effective supernet training in this scope. Moreover, the great idea about reducing parameter counts for **any SVD** is making the contribution a lot more useful in a general sense.

Hence, I decided to increase my score and vote for acceptance of the paper.

**Key Questions For Authors:**

- What are the inference benefits of using the proposed low rank compression with G from Sec. 3.1 (This part is very important to the significance of that idea)?
- How is the performance of individual architectures when not trained as long as in the proposed approach but scaled beyond a simple training? I.e. in Fig.7 the individual models are trained for the same *very* long time. Would they achieve similar performance with shorter training too? How is the cost - accuracy gained rate compared to your approach?
- DataSVD (SVD-LLM) with your search seems to outperform ACIP without fine-tuning. Is your search alone already better than ACIP's in a one to one comparison? If so, why? In their paper the numbers - for llama3.1 8B in particular seem to be much higher than the purple line in your plots would suggest.
- What happens if you compare to other approaches when they are using their own search like DOBI-SVD (without its remapping)?

**Limitations:**

yes

**Strengths And Weaknesses:**

## Soundness
### Strengest
- Broad evaluation and great performance on LLM, ViT-L, and DINOv3.
- Motivation for specific method for joint training is *extremely* well backed, both in theory and empirically.
- Demonstrating the performance of individually trained GPT-2 models against the FlexRank approach (Fig. 8) shows that improvements are also there at scale, which is reassuring.

### Major Weaknesses
- I very much appreciate the limitation discussion in the content. However, the omission of the **VERY** large cost (6 Days on 16 A100s) of running the approach should have been discussed there and not deferred to the supplement, especially considering that all other approaches are **significantly** less expensive.
- The claim of "consistently outperforms state-of-the-art compression" (l.69) in the third contribution is not backed by experiments. While the performance does indeed show improvements over other SVD methods, no comparisons to structured or semi-structured sparsity are presented, therefore not demonstrating superior performance over state-of-the-art compression in general.
- The comparison in the main Figure 4 does not seem entirely fair: ACIP was never meant to be used with joint LORA adapters. Still, authors find that "the results for ACIP in Fig. 4 show that adding shared trainable parameters to recover the compression error not only does not result in a meaningful improvement, but can be detrimental for recovering the baseline performance at full budget." (l.303). This is an interesting insight, but I would love to understand the performance  AND cost of the approach when used with its original setup - even if that requires per compression rate training.

### Minor Weakness
- "To the best of our knowledge, FLEXRANK is the first work that attempts to design flexible models at scale by operating at the factorized space." L.405. This is not entirely accurate. Specifically BitStack [1] recently introduced a flexible low rank based compression technique that can be dynamically adjusted in very fine grain to new memory constraints. Also FLORA [2] uses a supernet-like training scheme that allows flexible rank deployment after an initial training. However, the omission of the two is not diminishing the contribution of this work, only the accuracy of this particular statement.
- "enabling more adaptive deployment across diverse hardware and workloads." L.437 seems like a generalization assumption of the memory efficiency of the approach as no actual hardware measurements are provided. Showcasing throughput results in general would be good to also validate that the theoretical FLOPs improvements of Sec. 3.5.
- Tab 2: SVD-LLMv2 is not only applied to attention, the statement regarding this in the table is inaccurate.


## Presentation
### Major weaknesses
- The current introduction of "DataSVD" (Sec.3.1) is questionable to me. While there is a reference to the DRONE after the whole thing was introduced, the actual approach laid out in C.1. is just SVD-LLM/DRONE without fine-tuning. Not properly citing it in either of the two locations even though neither the minimization objective, nor the whitened SVD was introduced by this work is negligent. The statement "Data-aware decompositions of this form have been considered in prior work" in my opinion does not at all make it clear enough that **the exact formulation used here was introduced by prior work**.
- While the search is a critical starting point to reduce the Pareto front to a couple of candidates, reproducibility of the search component is difficult based on the provided information. Section C2 does provide some information, but no concrete information about (a.) the sensitivity criteria, (b.) the budgets used and (c.) the specific search formulation/algorithm including the $m_{k−1} \leq m_k$ constraint is provided.
- The notation is overly complicated and makes it harder to read the paper than would be necessary. E.g. in Sec.2.1 the expected cost should be below the budget $\beta$, but how is the hardware cost a function of expectation? Maybe the model performance $R$ is, but that is not even in the equation (l.93). Similarly, $\mathcal{B}$ is introduced as a budget in general but somehow shifts to Quantization, which is not even discussed in the main content. When moving to section 4, every section writes its objective differently, making it quite more difficult to read than would be necessary. PTS starts with just stating a loss, ASL then just calls it objective and is done and NL also calls it objective, but now defines M_NSL and a full argmin. Streamlining the formulations would make it significantly easier to read. More consideration whether introducing something actually improves understanding and removing less relevant parts would significantly aid the accessibility of the work.

### Minor weaknesses
- An algorithmic description of the final approach would greatly improve the understandability and reproducibility of the method presented.
- Figures
   - Using the same markers for all lines (difference is color only) in plots (Figure 2 is a nice exception) makes it harder to understand it for people with visual impairment.
   - quality of figure 3 & 8 is not great and could be improved by including it as vector figure.
- The sentence "Consequently, existing approaches typically either (i) optimize only the full model and then extract sub-models from frozen parameters, or (ii) [...]" (l.150) should have citations.

Spelling and other small mistakes
- l.117 "We make the key approximation that [...]" using *assumption* seems more appropriate than *approximation*.
- l.128 "Second, assuming approximate additivity of approximation errors across layers, [...]" approximate ... approximation (not so nicely written).
- l.242: "We also use for the controlled experiments " (grammar)
- l.289 "Among those, we compare with ACIP, the current state-of-the-art approach for low-rank elastic models." No citation for ACIP. Also, the cited reference is from arXiv, when it already has been accepted to TMLR.
- l.339 broken reference.
- l.284 DP is not introduced.



## Significance
Running LLMs in many targets without considerable extra overhead is a highly relevant problem. To me the significance of this work hinges on two components: 1.) The reformulation of the low rank forward as proposed in 3.5. could proof *extremely* relevant to all SVD compression - if it can be demonstrated that the theoretical FLOPs improvements translate to actual speedups. 2.) Authors clearly demonstrate that it is feasible to train multiple models at once using their dedicated training algorithm. However, the extremely high cost of training (according to D.4. it roughly scales linearly) seems prohibitive to use with larger models like Llama-3.1-70B where with the linear interpolation it would be around 60 days on 16 GPUs, limiting its applicability. Since 1.) has not been demonstrated and the cost of 2.) is very high compared to normal fine-tuning, without further clarification, the significance is low.

## Originality
Good Originality. First, the reformulation of the decomposition using the G matrix is a very cool idea that I have not seen anywhere despite its simplicity - beautiful (given that it actually improves inference). The training of the nested setup takes inspiration from supernet approaches and is not the first to do this on low rank, but is thoroughly investigated and distinct from anything similar that I am aware of.

[1] WANG, Xinghao, et al. BitStack: Any-size compression of large language models in variable memory environments. The Thirteenth International Conference on Learning Representations, 2025.

[2] CHANG, Chi-Chih, et al. Flora: Fine-grained low-rank architecture search for vision transformer. In: Proceedings of the IEEE/CVF Winter Conference on Applications of Computer Vision. 2024. S. 2482-2491.

---

> ### Author Rebuttal · Authors · 2026-03-31
>
> ## Soundness
>
> **Sound-W1: About reporting training cost.**
> We will move the discussion of training costs to the main paper. We also clarify that the reported wall-clock is not due to a fundamentally different objective: algorithmically, FlexRank training is comparable to standard distillation, while our current implementation is not yet optimized.
>
> **Sound-W2: About statement in l.69.**
> We agree that the statement in l.69 is too broad. Our claim should be restricted to **low-rank-based approaches**, and we will revise the wording accordingly.
>
> **Sound-W3: About comparison with ACIP.**
> ACIP uses LoRA adapters to compensate for compression after pruning an SVD-decomposed pretrained model with frozen base weights; it jointly optimizes adapters and pruning scores. In Fig. 4, ACIP denotes this original setup, while ACIP w/o LoRA removes the adapters to isolate the effect (see Sec. 5.1).
> We also note that ACIP was trained to convergence using the official code and early stopping; in our runs it stopped well before the maximum budget. While faster than FlexRank, it performs worse, consistent with our Sec. 4 analysis, since its objective resembles ASL.
>
> **Sound-MW1/MW3: About other works.**
> Thank you for pointing out BitStack and FLORA; we will include them in related work. We also appreciate the correction on SVD-LLMv2.
>
> **Sound-MW2: About memory efficiency at inference.**
> The claimed memory/FLOP savings follow directly from Sec. 3.5: a rank-$r$ forward uses exactly $(m+n-r)r$ parameters and multiplications. Demonstrating hardware speedups, however, requires an optimized low-level implementation of the factorized forward; we will clarify this distinction.
>
> ## Presentation
>
> **PW1: Citing DRONE.**
> We acknowledge the concern regarding attribution of the initialization stage. The detailed presentation was meant for completeness, not to claim novelty. We will cite DRONE more explicitly.
>
> **PW2: Details on the DP algorithm.**
> The sensitivity criterion $\mathcal{R}_{l,k}$ is the loss on a single batch induced by truncating layer $l$ to rank $k=p\cdot r$, with $p\in{0.1,0.2,\dots,1}$ and $r$ the maximum rank. In practice, we use 10 values of $p$, and increasing this resolution did not change the results. We will include pseudocode and release code for full reproducibility.
>
> **PW3: About notation.**
> The notation was intentionally broader than strictly necessary, to formalize elasticity and Pareto-elasticity beyond the low-rank case; for example, the hardware-cost term is written as an expectation over data. We agree this generality is not needed here and will simplify the notation accordingly.
>
> **Other minor weaknesses.**
> We will add an end-to-end algorithmic description, improve figure accessibility, fix typos, and revise citations as suggested.
>
> ## Significance
>
> **Sign-W1: Exercising inference speedup.**
> We agree that the speedup argument in Sec. 3.5 is general to SVD-style compression. Realizing it on hardware, however, requires a fused low-rank kernel: a high-level implementation of $(XA)B$ launches two kernels and materializes the intermediate activation, which can make the operation memory-bound and hide the theoretical gain. With kernel fusion, FLOP savings yield speedups.
>
>
> **Sign-W2: Clarification about training cost.**
> Algorithmically, each FlexRank training step is comparable to standard distillation, but with fewer total tokens and an elastic model. The current wall-clock is mainly due to implementation and memory overhead. In particular, our sliced low-rank layers prevent compilation-based optimization; without such optimizations, LoRA-style layers can be up to $4\times$ slower than dense ones at high rank.
> Training is also memory-bound: decomposed layers use roughly $2\times$ as much memory as dense layers, and distillation requires storing the teacher model; our hardware also has 64GB rather than 80GB A100s. With 80GB GPUs, we found that the per-GPU batch size can be doubled, halving gradient accumulation and yielding about $2\times$ speedup.
>
> **Q1: Inference advantage of Sec. 3.5.**
> The method reduces the forward cost from $(m+n)r$ to $(m+n-r)r < mn$ for any $r<\min(m,n)$, while keeping the operation structured.
>
> **Q2.**
> We believe the reviewer refers to Fig. 8, where we compare FlexRank to independently training each submodel for the full budget. This shows that joint optimization is crucial for recovering the nested structure needed for elastic deployment.
>
> **Q3.**
> ACIP uses vanilla SVD rather than DataSVD, so the relevant comparison in Fig. 4 is between the purple and green curves. There, our search is better on Llama-3.2-1B, comparable on Llama-3.1-8B, and worse on Llama-3.2-3B; thus we do not claim it is always superior, especially when the initialization is weaker.
>
> **Q4.**
> We will include DOBI-SVD in the revision. More generally, other search procedures that preserve global nestedness could be combined into FlexRank initialization. Still, we do not expect large gains from search alone.

---

> > ### Author Rebuttal · Reviewer_WDVJ · 2026-04-02
> >
> > **I appreciate the clarifications** on method specifics and the promise of code release for improved reproducibility. Moreover, once the training is done, both the elasticity and the overall performance are good. Also thanks for clarifying on the DataSVD search impact.
> >
> > However, **currently, I am not willing to change my score**.
> >
> > Even with further acceleration, the presented method remains very expensive compared to the methods it is compared against. ACIP in particular claims in their work that their algorithm takes less than 30 mins to run on a single GPU. Even when considering aggressive acceleration according to your assumptions (2x4x2x=16x) the remaining cost of 12 GPU days (8 nodes x 4 GPUs each x6 days / 16x acceleration, taken from section D.4) is hardly comparable to the other works in the main comparison figure and limits scaling ability to larger networks more commonly used. Hence, in my opinion, the **significance remains limited**.
> >
> > For the **soundness part**, given the large execution time differences, the comparison in Fig.4 is hardly fair, making it difficult to assess the concrete performance gains of your method. While Fig.8 specifically investigates individually trained vs. flex rank trained performance differences for GPT-2, a comparison to individually trained models should be present in at least one of the plots in Fig.4, where, currently, the comparison seems to be focused on comparing with mostly non-trained or only lora adjusted counterparts. This is going in a similar direction to Hay1's point about evaluating with **matched budgets and tuning effort**.
> >
> > Consequently, I have two more questions that I would appreciate clarification on:
> > 1. **Have you trained e.g. a DataSVDed Llama model individually in a training setting comparable to your elastic LLM (minus elasticity of cause)?** If so how does it compare to the other approaches in Fig.4? E.g. DataSVD seems very strong even without any training. Is it matching or surpassing your Flex network there?
> > 2. The parameter count comparison matters. While you introduced the reduction of parameter count for SVD-based compression, it can be applied to all of them. As I mentioned in my review, I love the idea! My concern is that its impact and the impact of the rest of your approach are conflated, preventing a fair apples-to-apples comparison between your search/training and effectiveness of other methods. Hence, my question is: **did you apply it the same (SVD general) parameter reduction to all SVD methods in Fig.4?** If not, the data points of the others would shift by around 0.1 to the left (in the 0.7 to 0.9 regime), wouldn't they? If that is the case, I would appreciate an update to the plot such that is a direct comparison and an additional comparison (maybe in the supplement) that explicitly shows parameter count vs accuracy with and without your idea.
> >
> > Getting clear answers on the two points above would significantly strengthen the comparison and enable comparing individual axis of your contributions to other works properly.

---

> > > ### Author Response · Authors · 2026-04-03
> > >
> > > Thank you for the follow-up. Before answering the questions, we would like to comment on the comparison with respect to wall-clock time to other approaches, since we understand this raises doubts about significance.
> > >
> > > Once it is established that our algorithm has **the same complexity as standard distillation** and once **the hardware infrastructure is fixed**, what matters for wall-clock time is the token budget. We do agree that ACIP will always be faster, but we have discussed on the issues of the approach: one of the outcome of our research is realizing that, if one wants to have submodels within a single set of weights that perform best at each budget (i.e. what we defined as “pareto-elastic models”), some non-trivial training is the price to pay, i.e. this does not emerge without training for it.
> > >
> > > This complements findings in the literature that adapters are often sufficient to recover from compression error in a single model and challenges the belief that such a simple approach could be as effective for elastic models. Moreover, outside low-rank training approaches, this seems to be supported by other works such as LlamaFlex (which is closed-source), which trains for 60B tokens (**$12\times$ our 5B budget**) and LayerSkip (included in our latest figure), which trains on $839B$ tokens (**$167\times$ our budget**). Consequently, if framed within approaches that can be competitive with a fully specialized model, it is evident that **FlexRank is cheaper than others**.
> > >
> > > In the context of model elasticity, training time is often less constraining than the ability to instantiate smaller models without retraining from scratch [2], as long as training the elastic model is comparable in terms of training effort [1].
> > > Given that the Llama3.2-1B has been trained on an engineered corpus of $9T$ tokens from the pretrained 8B version, methods of similar complexity to FlexRank are appealing and, indeed, significant for inference efficiency.
> > >
> > > Regarding soundness, we agree that a fairer comparison with lightweight approaches is difficult, even though we ensure we consider them at their best tuning/budget. However, in our latest figure, we compare to LayerSkip (thanks to publicly released weights), which is much heavier to train than ours: since limited by other approaches being not reproducible (no code or public weights), we believe that including LayerSkip should dispel doubts about our effort in guaranteeing fair comparison.
> > >
> > >
> > > **Q1: Comparison to individually trained models at matched budget**\
> > > We incorporated the reviewer's suggestion and compared it with individually trained models (after DataSVD).  In our [updated figure](https://ibb.co/qMJ29ZLF), we include training each of the 10 submodels, using 10% of the FlexRank training budget each, so that the total budget matches. In this scenario, FlexRank actually slightly outperforms the individually trained models on average; we hypothesize that this is because it leverages weight sharing, enabling smaller submodels to help optimize larger ones.
> > >
> > >
> > > **Q2: About parameter count comparison**\
> > > Thanks for the detailed read and for pointing this out; this allows us to clarify that **the technique in sec. 3.5 has already been applied to all rank-based approaches**, so the comparison faithfully reflects the relative performance of the methods. Therefore, its impact mainly affects comparisons between rank-based and non-rank-based approaches. We assumed it was clear that we do not consider the technique in Sec. 3.5 exclusively for FlexRank; we will mark it explicitly in the main text.
> > >
> > >
> > > [1] Han et al. Once-For-All: Train One Network and Specialize it for Efficient Deployment on Diverse Hardware, ICLR 2020 \
> > > [2] Yu et al., Slimmable Neural Networks, ICLR 2019

---

### Official Review · Reviewer_9Jp5 · 2026-02-25

**Soundness:** 3
**Presentation:** 3
**Significance:** 3
**Originality:** 3
**Overall Recommendation:** 5
**Confidence:** 3

**Summary:**

FLEXRANK leverages low-rank weight decomposition with nested, importance-based consolidation to extract submodels of increasing capabilities. It enables a “train-once, deployeverywhere” paradigm that offers a graceful tradeoff between cost and performance without training from scratch for each budget.

**Compliance With Llm Reviewing Policy:**

Affirmed.

**Final Justification:**

The rebuttal addressed my concerns. I will raise my score.

**Key Questions For Authors:**

Please refer to the weakness.

**Limitations:**

yes

**Strengths And Weaknesses:**

# Strength
+ The paper is well organized and the problem is well defined.
+ Comprehensive experimental results and theoretical insight.

# Weakness
+ Why is the curve in figure 4&5 so smooth? The number of parameters after SVD should be doubled. For example, even with 1.6 times the number of parameters after SVD, the performance should still be inferior to the original model, because only 0.8 times the original parameters are used.
+ Lack of comparison with pruning and quantization methods, as well as methods that predetermine a small set of submodel sizes.

---

> ### Author Rebuttal · Authors · 2026-03-31
>
> **W1: Regarding the parameter count for SVD-based methods**
> We confirm that the parameter count in Figures 4-5 is correct, and the reason why it does not double after SVD is related to the post-processing step we detail in section 3.5. In practice, once a target rank $r$ is fixed, by leveraging the non-uniqueness of the factorization, one can express $UV^\top=(UG)(VG^{-1})^\top$ and choose G such that the top $r\times r$ block of $\tilde{U}=UG$ is the identity, hence that block does not need to be stored. The matrix $\tilde{V}=(VG^{-1}) \in \mathbb{R}^{r \times n}$ is instead stored as usual. In summary, the parameter count adds to a total of $(m+n-r)\times r < m\times n$ for any $r < \min(m,n)$.
>
> **W2: Additional comparison with other compression families**
> Please refer to the [attached figure](https://ibb.co/20QMtDVF), where we provide additional results for LLM-Pruner, as a representative method of structured pruning, and LayerSkip, which allows early exit at intermediate layers of transformer block, thus producing a predetermined small set of submodel sizes. \
> Regarding quantization, we point out that it is an orthogonal approach to our method, and so we provide results combining FlexRank with post-train quantization (i.e. applied directly after FlexRank). For quantization, we converted all the 32-bits linear layers (excluding the lm_head) into 4-bits linear layers using bitsandbytes.
>
> As it is possible to notice from results on Llama3.2-1B, FlexRank consistently outperforms those methods, proving low-rank compression a practical and effective way for model elasticity.

---

> > ### Author Rebuttal · Reviewer_9Jp5 · 2026-04-02
> >
> > The rebuttal addressed my concerns. I will raise my score.

---

### Official Review · Reviewer_Hay1 · 2026-03-04

**Soundness:** 2
**Presentation:** 3
**Significance:** 2
**Originality:** 2
**Overall Recommendation:** 3
**Confidence:** 3

**Summary:**

This paper studies elastic deployment from a single pretrained model and proposes FlexRank: it builds a shared low-rank parameterization, uses DP search to obtain nested rank configurations, and then jointly trains submodels via distillation. The problem is practically relevant, and experiments span NLP/CV models at multiple scales.

**Compliance With Llm Reviewing Policy:**

Affirmed.

**Final Justification:**

Considering the detailed rebuttal from the authors and the paper, including the context from the rebuttal in the revised version would be good. I still hold my original score, given the paper's main text. But it is acceptable for me if this paper gets accepted.

**Key Questions For Authors:**

1. How can you rigorously justify (beyond empirical observation) the layer-wise additivity assumption used in the DP search for deep nonlinear models? Could you provide a quantitative analysis linking approximation error to downstream performance?

2. For comparisons with ACIP/MatFormer/LlamaFlex, could you report fully matched training and tuning budgets (GPU-days, training steps, hyperparameter search space) together with convergence curves, to rule out gains from unequal optimization effort?

**Limitations:**

yes

**Strengths And Weaknesses:**

Strengths
1. Clear and practical objective: train once and deploy submodels under different budgets.
2. End-to-end engineering pipeline: DataSVD initialization + configuration search + distillation training.
3. Broad empirical coverage across architectures, with relatively smooth degradation under tighter budgets.

Main Concerns
1. Limited novelty: nested/elastic submodel deployment is not new; the paper is closer to an integration and scaling of existing ideas than a fundamentally new direction [C1-C3].
2. Low-rank compression itself is also a mature line of work; methodological novelty on this axis is limited [C4].
3. The DP selection relies on a strong layer-wise additivity / weak-coupling assumption, which looks more like a practical heuristic than a substantial algorithmic innovation.
4. Comparison fairness can be improved: the paper lacks stricter matched-budget / matched-tuning / matched-convergence reporting.
5. Training cost is high (Appendix D.4), but there is no clear cost-performance curve against major baselines under matched resources.


References

[C1] Once-for-All (OFA), ICLR 2020. https://arxiv.org/abs/1908.09791

[C2] MatFormer: Nested Transformer for Elastic Inference, NeurIPS 2024. https://papers.nips.cc/paper_files/paper/2024/hash/fe066022bab2a6c6a3c57032a1623c70-Abstract-Conference.html

[C3] LlamaFlex: Many-in-One LLMs via Generalized Pruning and Weight Sharing, ICLR 2025. https://openreview.net/forum?id=AyC4uxx2HW

[C4] DRONE: Data-aware Low-Rank Compression for Large NLP Models, NeurIPS 2021. https://proceedings.neurips.cc/paper/2021/hash/f56de5ef149cf0aedcc8f4797031e229-Abstract.html

---

> ### Author Rebuttal · Authors · 2026-03-31
>
> **W1-W2: About the contributions of our work:** \
> Thank you for your comments. We agree that elastic deployment and low-rank compression are established areas, and our goal is not to introduce these paradigms, but to resolve a previously unaddressed problem within their intersection: how to train a single low-rank parameterization that is simultaneously optimal across multiple deployment budgets. Therefore, our contribution is not a mere combination of existing components, but a principled characterization and solution of elastic training in rank space, with two key novel elements:
> (i) Theoretical insight: We show that two natural alternatives, optimizing only the full model and post-selecting submodels, or jointly optimizing arbitrary submodels, do not recover Pareto-optimal reduced-rank submodels even in a simplified linear setting; this isolates nestedness as the key structural requirement for elastic low-rank training;
> (ii) Algorithmic realization: We propose a practical pipeline (data-aware initialization based on the same data-aware decomposition idea as [C4], DP-based nested rank selection, and joint distillation) that achieves substantially better accuracy–budget trade-offs than prior low-rank elastic baselines across LLMs and ViTs.
> Importantly, prior works such as [C1-C3] explore elasticity along different axes (e.g., depth, width, pruning), while [C4] addresses low-rank compression without elasticity. Our work is the first to systematically address elasticity within the low-rank regime, combining a structural characterization with an effective training procedure.
>
> **W3-Q1: About the assumptions behind the DP algorithm:**\
> We agree that the additivity assumption is an approximation rather than an exact property of deep nonlinear models. Its role is to make the global rank-allocation search tractable: without additional structure, jointly selecting per-layer truncations under a budget is combinatorial and infeasible at realistic scale.
> This is indeed why we proposed the exhaustive experimentation on MNIST, that already quantitatively shows that our search finds the best performing submodels. We acknowledge that the scope remains limited to that setting, but due to the nature of the problem, it is not feasible to scale this analysis to larger networks.
>
> **W4:  About fair comparison** \
> Regarding reporting the results at matched training budget, we note that all the baselines are reported at their convergence, given that SVD/DataSVD are not gradient-based and ACIP runs based on an early-stopping criterion. Moreover DataSVD is the initialization of FlexRank, so at matched budget they coincide. Regarding ACIP, as we discuss in the response to W5, it trains much less but does not benefit from more training.  To address this more rigorously and avoid bias, we will report this in the manuscript and will include comparisons across matched training budgets.
>
> **Q2: Additional details on hyperparameters and training budget** \
> To clarify the comparison protocol, we aimed to evaluate each baseline fairly, using the official implementation when available and a nontrivial tuning budget, rather than allocating extra optimization effort in favor of our method. In particular, for ACIP we used the official code, which includes an early stopping criterion over a potentially large number of optimization steps; in our runs, training typically stopped well before the maximum allowed budget. We will report full details in the revision due to limits in rebuttal length.
>
> **W5: Regarding the training cost details** \
> We agree that the current wall-clock time is high, and we clarify here both the sources of overhead and what would be required to reduce it in practice.Algorithmically, FlexRank has the same per-step complexity as standard distillation training, yet three implementation-level factors currently increase latency:
> (i) Dynamic rank sampling prevents compilation. During training, each step samples different submodel ranks, which changes tensor shapes and prevents the use of torch.compile or other graph-level optimizations. A potential solution would include static-shape execution for compilation via masking on static CUDA graphs and CUDA structured sparsity kernels.
> (ii) Two-model for distillation. Training requires both the teacher and student, doubling forward-pass cost and increasing memory pressure, limiting batch size and throughput. One could disaggregate model hosting and allow for asynchronous rollouts through vllm/sglang.
> (iii) Parameterization overhead. The factorized layers introduce additional memory movement (e.g., intermediate activations in $XA$), making the implementation more memory-bound than dense layers when not fused. Implementing the factorized forward pass as a single fused kernel could recover the memory bottleneck. While valuable, these optimizations fall outside of the immediate scope of our study. We would be happy to move the performance section and above commentary in the main manuscript.

---

> > ### Author Rebuttal · Reviewer_Hay1 · 2026-04-02
> >
> > Thank you for the rebuttal.
> >
> > The response helps clarify some points, but my main concerns remain only partially addressed. In particular, I still do not find the justification for the DP assumption sufficiently convincing, and the fairness of the comparisons is still not fully established under matched budgets and tuning effort.
> >
> > Therefore, I will maintain my original score.

---

> > > ### Author Response · Authors · 2026-04-03
> > >
> > > Thank you for the follow-up. We understand that your remaining concerns are centered on (i) the DP assumption and (ii) whether extra optimization effort could explain the gains.
> > > ## In-depth analysis of ranking-preserving assumption
> > > The DP assumption should not be interpreted as an exact theorem/solution: the role of the additive probe is to make the otherwise combinatorial rank-allocation problem tractable. **What is needed to preserve optimal candidates is not the exact additivity, but that the probe induces a sufficiently reliable ordering over candidate solutions**, which is substantially weaker. This is also how we frame the method in the paper. The question, therefore, becomes whether this “ranking-preservation” is reasonable in practice. Our view is that, while it cannot be tractably shown in full generality for arbitrary deep nonlinear networks, it is reasonable in the regimes we study. We provide the following analysis below:
> > >
> > > **Notation**\
> > > Call $m=(m_1, \dots, m_L)$ any submodel induced by a rank-selection over $L$ layers, and $s(m)=(s_{m_1}, \dots, s_{m_l})$, where $s_{m_l}$ is the model’s sensitivity to rank reduction at each layer independently. Define  $A(m):=\sum_{l=1}^{L} s_{m_l}$ the additive probe used by DP, and $F(m)$ the true joint probing loss, obtained by directly testing model $m$. The experiment is conducted on MNIST on a search space of $10^4$ submodels (same setting as our Figure 3).
> > >
> > > **Metrics**
> > > - $\rho \in [-1,1]$: it is the Spearman's correlation between $A(m)$ and $F(m)$; a value close to 1 indicates stronger ranking agreement
> > > - $v \in [0,1]$: it is the pairwise violation rate, which measures the ranking-inversions; a value closer to 0 indicates fewer inversions.
> > > - $p \in [0,1]$: it is the fraction of cases in which the optimal model found by DP is also optimal w.r.t. $F(m)$, higher is better.
> > > - $r_{avg},r_{max}$, relative to min eval loss: it is the margin between the $A(m)$ and $F(m)$ when DP does not choose the best model w.r.t. $F(m)$, lower is better.
> > >
> > > **Discussion**\
> > > Results presented in [our analysis](https://ibb.co/Mx6ZsH59) show:
> > > - **Very high global-level rank-agreement (Fig. A):** Each point is one submodel, and the plot shows the submodel’s position in the global ranking, i.e., a point $(x,y)$ means “this submodel is in the top $x\\%$ for $A(m)$ and in the top $y\\%$ for $F(m)$”. Values near the diagonal mean that $A(m)$ and $F(m)$ give that submodel a similar global ranking. The Spearman coefficient $\rho=0.991$ and violation rate $v=3.7\\%$ (tab. D) indicate very strong global monotone agreement and very low global ranking error.
> > > - **High DP success rate and low regret:** DP matches the true exact-budget winner on `p=94.1%` of cases over all the front of best submodels (Fig. B); more importantly, when it does not return the best submodel, the regret is low. Figure C shows the empirical CDF of the resulting relative regret: for any threshold on the $x$-axis, the $y$-value gives the fraction of DP failures whose regret is at most that threshold. The fact that this curve rises quickly in the low-regret region and saturates before 12% shows that most failures remain close to the true exact-budget optimum.
> > > In practice, DP often finds the same solution we would have found by exhaustive exploration, and when it does not, the solution it finds is roughly equivalent in terms of loss.
> > >
> > > **Closing remarks.**\
> > > The DP algorithm is only one way to search for submodels, and we have shown that it performs reasonably well in practice. However, the search is not by itself our main contribution: our paper is about properly training elastic networks in the low-rank parametrization.
> > >
> > > ---
> > >
> > > ## Regarding comparisons under matched budgets and tuning effort
> > > In our [new comparison](https://ibb.co/qMJ29ZLF), we included a baseline that trains the same submodels selected for FlexRank, but in isolation and matching the overall training budget. This is arguably a strong baseline, since we are comparing it to a non-elastic method (i.e., 10 different sets of weights). The fact that FlexRank slightly outperforms it on average indicates that its training objective effectively leverages weight sharing, enabling smaller submodels to optimize larger ones.
> > >
> > > Regarding fair comparison, we compared methods that are publicly reproducible (public code or weights), and we have considered them at their best tuning/budget. This holds both for lightweight ones (like LLM-Pruner and ACIP) and for methods that are much heavier than FlexRank, such as LayerSkip, which has similar complexity to FlexRank but has been trained on 839B tokens (i.e., $167\times$ our 5B tokens budget).
> > >
> > > For further context, similarly complex approaches such as LLamaFlex [C3] train on 60B tokens ($12\times$ ours), while the standard recipe for obtaining the Llama3.2-1B model from the pretrained 8B version trains on 9T tokens [1] ($1800\times$ ours).
> > >
> > > [1] [LLama3.2-1B model card on HuggingFace](https://huggingface.co/meta-llama/Llama-3.2-1B)

---

### Official Review · Reviewer_qqeX · 2026-03-05

**Soundness:** 3
**Presentation:** 2
**Significance:** 3
**Originality:** 4
**Overall Recommendation:** 4
**Confidence:** 3

**Summary:**

The authors study the problem of compressing a pre-trained deep neural network into a family of smaller models spanning multiple parameter budgets. They propose a method that, given a trained model and a reference dataset, learns a single set of elastic parameters that can be instantiated at different sizes. At deployment time, the same parameterization can be “shrunk” to meet a target budget, yielding a favorable trade-off between parameter count and predictive performance across the resulting model family.

The compression method relies on two main ingredients: (i) the parameter sets for different target sizes are nested, so each larger model contains the smaller ones as a subset, and (ii) the shared low-rank factors are trained jointly across all compression levels using a distillation objective against the original pretrained model. In experiments on Llama- and ViT-style models, this approach substantially outperforms elastic baselines based on SVD-style weight decompositions in the reported setting. Moreover, the authors provide theoretical results for linear networks suggesting that joint optimization over nested parameter sets is necessary to recover the Pareto-optimal trade-off across budgets.

**Compliance With Llm Reviewing Policy:**

Affirmed.

**Final Justification:**

The paper presents a novel compression framework that provides a useful foundation for future work. The empirical validation is strong, and the authors also strengthened the submission by including additional baseline comparisons in the rebuttal. My decision is a weak accept because the Pareto-optimal framing appears to hold only within the relatively narrow setting of low-rank compression regimes. Although the authors indicate that they will refine this positioning in the final version, I believe that doing so will also make the Pareto-front result somewhat less impactful. Overall, I view the paper as a meaningful contribution to the literature and support its publication, but I do not yet find parts of the Pareto narrative sufficiently developed to justify a stronger accept.

**Key Questions For Authors:**

Why did the authors choose to only compare their method against rank-approximation based compression techniques? Do you find similar performance gains over SotA when considering a diverse set of compression techniques?

**Limitations:**

yes

**Strengths And Weaknesses:**

Strengths
- Original and effective method. The overall procedure is easy to understand and performs well in practice. The idea of jointly minimizing reconstruction error across nested compression levels is well motivated, and experiments show clear gains over prior approaches.
- Strong empirical trade-offs. Empiric evaluations indicate improved accuracy/parameter budget trade-offs compared to SVD-based baselines. A qualitative example suggests the learned importance scores are meaningful.
- Theory is targeted and relevant and aligns with the empirical message. The theoretical analysis is sound and supports the main claim: in a simplified surrogate setting, optimizing all nested compression levels jointly is necessary for recovering a Pareto frontier across at all levels.
- Ablations support the main contribution. Ablations show that data-dependent training of the factorization substantially improves performance, but does not fully explain the gains; the joint optimization over nested levels contributes materially as well.

Weaknesses
- The scope of the Pareto-optimality claim seems overstated. Definition 2.1 frames a “Pareto elastic model” over the entire class of compression strategies $\mathcal T_\beta$, including methods beyond low-rank approximation (the paper mentions sparsification and quantization). However, the main experiment used to support the claim (“Recovering the True Pareto Front”) only evaluates low-rank approximation-based compression. So the results demonstrate Pareto-front recovery within the low-rank regime, but don’t justify the broader, strategy-agnostic Pareto claim implied by Definition 2.1. It would strengthen the paper to either (i) state the claim explicitly in the low-rank regime, or (ii) extend Section 3.4 to include a small set of representative non–low-rank compression families consistent with Definition 2.1.
- The empirical section only considers rank-approximation baselines. To me, Section 5 would be more convincing if it either (i) included a few non–rank-based elastic compression baselines (e.g., from the “flexible networks” line of work) to support the implicit claim that rank-based compression is the most appropriate regime here, or (ii) stated explicitly why such baselines are not applicable in this setting. As written, it remains unclear why SVD-style low-rank approximations are treated as the primary comparator.
- I found parts of the presentation hard to follow on the first pass. Some notation and phrasing is under-specified, and I had to reread several places. Examples include: line 158 (“Solving the objective via SVD”), Section 3.5 (unclear definitions of $m,n$), Section 4.1 (unclear definition of $M^\star$) and line 263 ("the chance that any algorithm that only operates on (9) would find global minimizer of (7) is zero" - "chance" with respect to what randomness). Figure 4 contains an unresolved reference.

Overall, the paper presents an original and effective approach for compressing a pretrained network into a single elastic model that can be instantiated across multiple parameter budgets. Theoretical analysis and empirical results both suggest that the proposed joint, nested training setup yields a meaningful performance advantage over previous methods. That said, the headline Pareto claim appears somewhat broader than what is directly supported by the current experiments, and the empirical comparisons focus on a relatively narrow set of baselines.

---

> ### Author Rebuttal · Authors · 2026-03-31
>
> **W1: About pareto-optimality definition and scope.** \
> Thank you for pointing this out. The definitions in Sec. 2 are intended as **general conceptual definitions** of model elasticity and Pareto-elasticity, motivated by the lack of a common formalization in prior work. Our goal there is to state clearly what properties an elastic model should satisfy, independently of the specific compression mechanism.
> That said, the guarantees and evidence provided in this paper are specific to the **low-rank compression regime**. In particular, the theoretical development in Sec. 4 and the “recovering the Pareto front” experiment in Sec. 3.4 concerns nested low-rank submodels rather than arbitrary compression families. We agree that the current wording can be read too broadly, and in the revision we will make the scope explicit throughout, replacing any wording that could be interpreted as claiming Pareto-optimality across compression strategies in general.
> To better contextualize the method beyond rank-based baselines, we also include additional comparisons in the [attached figure](https://ibb.co/20QMtDVF) with representative methods from other compression families: LLM-Pruner (structured pruning), LayerSkip (variable depth), and a simple post-training quantization variant applied on top of FlexRank.
>
>
> **W2: Clarifying the writing and presentation.** \
> Thank you. We will revise the presentation to make the notation and phrasing more precise throughout.
> Regarding the wording around Thm. 4.1, you are right that “chance” is imprecise. The theorem is a measure-theoretic statement: among the set of global minimizers of Eq. (9), the subset that also yields zero submodel optimality gap for rank $r<k$ has Lebesgue measure zero relative to that solution set. Intuitively, this means that a solution obtained by optimizing only the full model will almost never coincide with the special factorization structure needed to recover all reduced-rank Pareto-optimal submodels. We will revise the text to state this precisely and avoid informal probabilistic language.
>
> **W2-Q1: About comparison with other families of compression.** \
> Our goal in this work is to advance rank-based elastic compression, so prior methods in that regime are the most direct and informative baselines. We do not claim that low-rank compression is the only possible route to elasticity, nor that it universally dominates other compression axes such as pruning, depth elasticity, or quantization. Rather, we focus on low-rank compression because it provides a natural importance ordering through “SVD-like decompositions”, which in turn yields a principled route to nested submodel construction.
> These compression axes are also complementary rather than mutually exclusive. For example, one can combine low-rank compression with depth reduction or quantization. To provide broader context, in the [attached figure](https://ibb.co/20QMtDVF) we additionally compare against representative methods from other families, including LLM-Pruner (structured pruning), LayerSkip (variable depth), and a simple post-training quantization variant applied on top of FlexRank. On Llama-3.2-1B, these additional results suggest that FlexRank is competitive beyond rank-only comparisons, while also being compatible with quantization. We will add these results to better position the paper and clarify the scope of our claims.

---

> > ### Author Rebuttal · Reviewer_qqeX · 2026-04-03
> >
> > I will maintain my score.

---

### Decision · Program_Chairs · 2026-04-30

**Decision:**

Accept (spotlight)

**Comment:**

This paper introduces a new framework to decompose large pretrained models, such as Large Language Models (LLMs) and Vision Transformers (ViTs), into a family of nested low-rank submodels. To enable adaptive deployment, the method includes a data-aware low-rank initialization, a dynamic programming search for optimal nested rank selection, and a joint knowledge distillation phase. By extracting importance-ordered subnetworks that share a single set of parameters, this method enables a "train-once, deploy-everywhere" paradigm. This allows models to dynamically adapt to various computational and memory constraints, offering a flexible trade-off between deployment cost and task performance.

The reviewers commended the paper's original nested training approach, strong empirical performance, and theoretical insights. The authors successfully resolved several major concerns by explicitly narrowing their "Pareto-optimality" claims, clarifying parameter count calculations, and providing new comparisons. However, some issues remained only partially resolved: reviewers maintained reservations regarding the high training overhead compared to lightweight baselines, the absolute fairness of matched-budget evaluations, and the lack of rigorous theoretical justification for the dynamic programming part/assumptions. The authors are encouraged to integrate the expanded matched-budget baselines, explicitly discuss the wall-clock training cost limitations in the main text, and properly contextualize the dynamic programming heuristic and the theoretical claims.